# ROYAL SOCIETY
# OPEN SCIENCE

applied mathematics/differential equations/
mathematical modelling

Gaussian process, parameter inference, nonlinear ordinary differential equations, partial observations

**Authors for correspondence:**
Huaxiong Huang
e-mail: hhuang@uic.edu.cn
Shixin Xu
e-mail: shixin.xu@dukekunshan.edu.cn

# Numerical method for parameter inference of systems of nonlinear ordinary differential equations with partial observations

Yu Chen[1,2], Jin Cheng[3,1], Arvind Gupta[4],
Huaxiong Huang[5,6,2,4] and Shixin Xu[7]

[1]School of Mathematics, Shanghai University of Finance and Economics, Shanghai, People's Republic of China
[2]Centre for Quantitative Analysis and Modeling (CQAM), The Fields Institute for Research in Mathematical Sciences, 222 College Street, Toronto, Ontario, Canada
[3]School of Mathematical Sciences, Fudan University, Shanghai 200433, People's Republic of China
[4]Computer Science, University of Toronto, Toronto, Ontario, Canada
[5]Joint Mathematical Research Centre of Beijing Normal University and BNU-HKBU United International College, Zhuhai, People's Republic of China
[6]Department of Mathematics and Statistics, York University, Toronto, Ontario, Canada
[7]Duke Kunshan University, 8 Duke Ave, Kunshan, Jiangsu, People's Republic of China

SX, 0000-0002-8207-7313

Parameter inference of dynamical systems is a challenging task faced by many researchers and practitioners across various fields. In many applications, it is common that only limited variables are observable. In this paper, we propose a method for parameter inference of a system of nonlinear coupled ordinary differential equations with partial observations. Our method combines fast Gaussian process-based gradient matching and deterministic optimization algorithms. By using initial values obtained by Bayesian steps with low sampling numbers, our deterministic optimization algorithm is both accurate, robust and efficient with partial observations and large noise.

## 1. Introduction

Many problems in science and engineering can be modelled by systems of ordinary differential equations (ODEs). It is often difficult or impossible to measure some parameters of the systems directly. Therefore, various methods have been developed to estimate parameters based on available data.

Mathematically, such problems are classified as inverse problems, which have been widely studied [1–4]. They can be also treated as *parameter inference* in statistics. Statistical inference enables the construction of data-efficient learning machines and shows advantages in dealing with noisy data, which is increasingly being studied and discussed (e.g. [5–7]).

For nonlinear ODEs, standard statistical inference is time consuming as numerical integration is needed after each update of the parameters [8,9]. Recently, gradient matching techniques have been proposed to circumvent the high computational cost of numerical integration [7,10–12]. These techniques are based on minimizing the difference between the time derivatives (gradients) and the right-hand side of the equations. This usually involves a process consisting of two steps: data interpolation and parameter adaptation. Among them, non-parametric Bayesian modelling with Gaussian processes is one of the promising approaches, which includes data interpolation by Gaussian process and parameter adaptation by matching the solution or model. An adaptive gradient matching method based on a product-of-experts approach and a marginalization over the derivatives of the state variables was proposed by Calderhead *et al.* [8] and extended by Dondelinger *et al.* [10]. Barber & Wang [13] proposed a Gaussian process-based method in which the state variables are marginalized. Macdonald *et al.* [9] provided an interpretation of the above paradigms. Wenk *et al.* [14] proposed a fast Gaussian process-based gradient matching (FGPGM) algorithm with theoretical framework in systems of nonlinear ODEs, which was indicated more accurate, robust and efficient.

For many practical problems, the variables are only *partially observable*, or not at all times [15]. As a consequence, parameter inference is more challenging, even for a coupled system where the parameters are uniquely determined by data of partially observed data under certain initial conditions. It is not clear whether the gradient matching techniques can be applied to the case when there are latent variables. The Markov chain Monte Carlo algorithm has the ability to sidestep the issue of parameter identifiability in many cases, but convergence remains a serious issue [7]. Therefore, the graphical model, density formula and algorithm for inference with partial observations, especially how to deal with the latent variables, are worth investigation. Moreover, we need to pay attention to the feasibility, accuracy, robustness and computational cost of numerical computations for such problems.

In this work, we focus on the case of parameter inference with partially observable data. The main idea is to treat the observable and non-observable variables differently. For observable variables, we use the same approach as proposed by Wenk *et al.* [14]. We will provide three approaches to deal with the unobserved variables, including numerical integrations and Gaussian process sampling, and compare their performances. To circumvent the high computational cost of sampling in Bayesian approaches, we also combine the extended FGPGM with least square optimization method. The remaining part of the paper is organized as follows. In §2, we give the numerical method to deal with parameter identification problems with partial observation. Numerical examples are presented in §3. Some concluding remarks are given in §4.

## 2. Algorithm

The main strategy of Gaussian process-based gradient matching is to minimize the mismatch between the data and the ODE solutions in a maximum-likelihood sense, making use of the property that Gaussian process is closed under differentiation. In this section, we will extend the FGPGM method proposed in [14] to the situation that contains unobserved variables.

In this work, we would like to estimate the time-independent parameters $\boldsymbol{\theta}$ of the following dynamical system described by

$$\dot{X}(t) = f(X(t), \boldsymbol{\theta}), \tag{2.1}$$

where $\dot{X}$ is the vector of time derivative of the variable $X = (X_1(t), X_2(t), \ldots, X_{N_1}(t))$ and $f$ can be nonlinear vector valued functions. We assume only part of the variables are measurable and denote them as $X_M$. Throughout the paper, we use subscript $M$ to specify measurable components. They are observed on discrete time points as $Y(t_i)(i = 1, \ldots N_2)$ with noise $\varepsilon$ such that $Y = X_M + \varepsilon$. We assume that the noise is Gaussian $\epsilon(t_i) \sim \mathcal{N}(\mathbf{0}, \sigma^2 I)$, then

$$\rho(\boldsymbol{y}|\boldsymbol{x}_M, \sigma) = \mathcal{N}(\boldsymbol{y}|\boldsymbol{x}_M, \sigma^2 I), \tag{2.2}$$

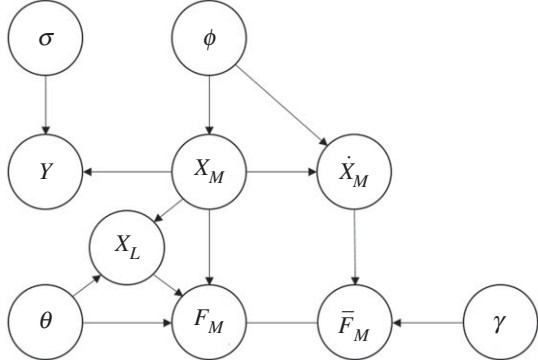

**Figure 1.** Chain graph with partial observable variables.

where $x_M$ and $y$ are the realizations of $X_M$ and $Y$, respectively. The latent/unmeasurable variables are denoted as $X_L$, with $\dim(X_M) + \dim(X_L) = \dim(X)$.

The idea of Gaussian process-based gradient matching is as follows. Firstly, we put a Gaussian process prior on $x_M$,

$$\rho(x_M|\boldsymbol{\mu}_M, \phi) = \mathcal{N}(x_M|\boldsymbol{\mu}_M, \boldsymbol{C}_\phi). \tag{2.3}$$

Here $\boldsymbol{C}_\phi$ is denoted as the covariance matrix. Its components are given by $\boldsymbol{C}_\phi(i, j) = k_\phi(t_i, t_j)$ with respect to a kernel function $k_\phi$ parametrized by the hyperparameters $\phi$. Then according to lemma A.8, the conditional distribution of the $k$th state derivatives is

$$\rho(\dot{x}_k|x_k, \phi_k) = \mathcal{N}(\dot{x}_k|\boldsymbol{D}_k x_k, \boldsymbol{A}_k), \tag{2.4}$$

where

$$\boldsymbol{D}_k = \boldsymbol{C}_{\phi_k}(\dot{x}_k, x_k)\boldsymbol{C}_{\phi_k}(x_k, x_k)^{-1}(x_k - \boldsymbol{\mu}_k) \tag{2.5}$$

and

$$\boldsymbol{A}_k = \boldsymbol{C}_{\phi_k}(\dot{x}_k, \dot{x}_k) - \boldsymbol{C}_{\phi_k}(\dot{x}_k, x_k)\boldsymbol{C}_{\phi_k}(x_k, x_k)^{-1}\boldsymbol{C}_{\phi_k}(x_k, \dot{x}_k). \tag{2.6}$$

Here $x_k = (x_k(t_1), x_k(t_2), \ldots, x_k(t_{N_2}))$. For more details, we refer to appendix A. We set up the following chain graph to show the relationship between the variables (figure 1). In figure 1, $F_M$ is the deterministic output of the measurable parts of the ODEs, whose realization is determined by $x$ and $\boldsymbol{\theta}$. $\bar{F}_M$ is introduced to incorporate the model uncertainty, which is assumed to be a Gaussian noise with standard deviation $\gamma$ so that for its realization $\bar{f}_M$ we have

$$\rho(\bar{f}_M|\dot{x}, \gamma) = \mathcal{N}(\bar{f}_M|\dot{x}, \gamma\boldsymbol{I}). \tag{2.7}$$

Then the joint density can be represented by the following theorem.

**Theorem 2.1.** *Given the modelling assumptions summarized in the graphical probabilistic model in figure 1,*

$$\begin{aligned} &\rho(x_M, \boldsymbol{\theta}|y, \boldsymbol{\phi}, \boldsymbol{\sigma}, \boldsymbol{\gamma}) \\ &\propto \rho(\boldsymbol{\theta})\mathcal{N}(x_M|\mathbf{0}, \boldsymbol{C}_\phi)\mathcal{N}(y|x_M, \sigma^2\boldsymbol{I})\mathcal{N}(f_M(x_M, \tilde{x}_L(x_M, \boldsymbol{\theta}), \boldsymbol{\theta})|\boldsymbol{D}x_M, \boldsymbol{A} + \gamma\boldsymbol{I}), \end{aligned} \tag{2.8}$$

*where $\tilde{x}_L(x_M, \boldsymbol{\theta})$ involved in $f_M$ is the solution determined by $\boldsymbol{\theta}$ and $x_M$.*

The proof is given in appendix B.

In our computation, $\tilde{x}_L$ can be obtained by integrating the ODE system numerically with proposed $\boldsymbol{\theta}$ and initial values of $x_M$ and $x_L$. Then, the target is to maximize the likelihood function $\rho(x_M, \boldsymbol{\theta}|y, \phi, \sigma, \gamma)$.

The present algorithm is a combination of a Gaussian process-based gradient matching and a least square optimization. In the GP gradient matching step, the Gaussian process model is first fitted by inferring the hyperparameter $\phi$. Secondly, the states and parameters are inferred using a one-chain MCMC scheme on the density as in [14]. Finally, the parameters estimated above are set as initial guess in the least square optimization. The algorithm can be summarized as follows.

## Algorithm

Input: $\boldsymbol{y}$, $\boldsymbol{f}(\boldsymbol{x}, \boldsymbol{\theta})$, $\gamma$, $N_{MCMC}$, $N_{burnin}$, $t$, $\sigma_s$, $\sigma_p$

Step 1. Fit Gaussian process model to data

Step 2. Infer $\boldsymbol{x}_M$, $\boldsymbol{x}_L$ and $\boldsymbol{\theta}$ using MCMC

$S \leftarrow \emptyset$

for $i = 1 \rightarrow N_{MCMC} + N_{burnin}$ do

 $\mathcal{T} \leftarrow \emptyset$

 for each state $x_{Mj}(t_k)$ do

 Propose a new state value using a Gaussian distribution with standard deviation $\sigma_s$

 Accept proposed value based on the density (equation (2.8))

 Add current value to $\mathcal{T}$

 end for

 for each parameter do

 Propose a new parameter value using a Gaussian distribution with standard deviation $\sigma_p$

 Infer $\boldsymbol{x}_L$ by integration or sampling (detailed in §2).

 Accept proposed value based on the density (equation (2.8))

 Add current value to $\mathcal{T}$

 end for

 Add the mean of $\mathcal{T}$ to $S$

end for

Discard the first $N_{burnin}$ samples of $S$

Return $\boldsymbol{x}_M$, $\boldsymbol{x}_L$, $\boldsymbol{\theta}$

Step 3. Optimization of equation (2.11) using $\boldsymbol{\theta}$ from Step 2 as initial guess.

In Step 1, the Gaussian process model is fitted to data by maximizing the log of marginal likelihood of the observations $\boldsymbol{y}$ at times $\boldsymbol{t}$

$$\log{(\rho(\boldsymbol{y}|\boldsymbol{t}, \phi, \sigma))} = -\frac{1}{2}\boldsymbol{y}^{\mathrm{T}}(\boldsymbol{C}_\phi + \sigma\boldsymbol{I})^{-1}\boldsymbol{y} - \frac{1}{2}\log{|\boldsymbol{C}_\phi + \sigma\boldsymbol{I}|} - \frac{n}{2}\log{2\pi}, \tag{2.9}$$

with respect to hyperparameters $\phi$ and $\sigma$. $\sigma$ is the estimated standard deviation of the observation noise and $n$ is the amount of observations.

For the inference of the unobserved variables, there are the following possible ways. The first method is to integrate the whole ODE system, which only depends on initial conditions and the proposed value of parameters. An alternative way is to integrate partially the system, that is, integrate the equations for the unobservable variables only. The coupled observable variables are regarded as known coefficients whose values are taken from the observations. In order to ensure the stability and convergence of the integration, smoothing and interpolation may be necessary if the data of the observable variables are sparse and noisy. The numerical integration of $\tilde{x}_L$ in these two methods are implemented only after each update of $\boldsymbol{\theta}$. The third approach to deal with the latent variables is applying the same sampling process as the observable states under the assumption of Gaussian process and doing the gradient matching, in which the posterior probability is adapted as

$$\rho(x_L|x_M, \boldsymbol{\theta}, \phi, \sigma, \gamma) = \rho(\boldsymbol{\theta})\mathcal{N}(x_L|\boldsymbol{0}, \boldsymbol{C}_\phi)\mathcal{N}(f_L(x_M, x_L, \boldsymbol{\theta})|Dx_L, \boldsymbol{A} + \gamma\boldsymbol{I}). \tag{2.10}$$

Note that although there is no observation to match for the latent variables, the gradient matching is still valid. Thus, in the integration approaches only the observable variables are sampled while for the third one all the variables are sampled together with the updates of the likelihood function in each MCMC

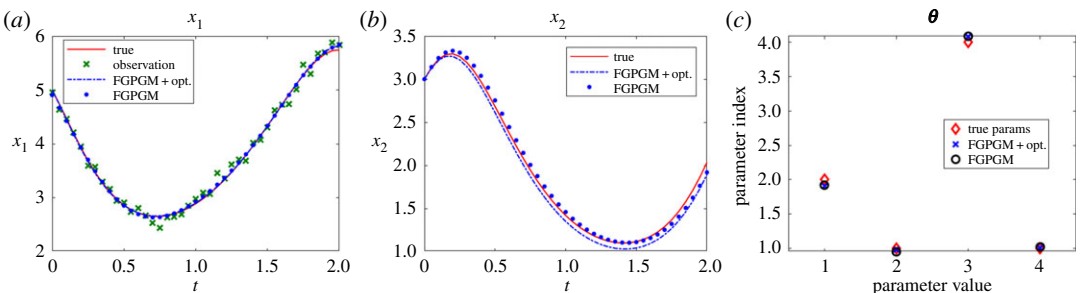

**Figure 2.** Reconstruction and inference results for the Lotka–Volterra system, showing the state evolution over time and parameter distributions. $x_1$ is observable and $x_2$ is latent variable. The ground truth, FGPGM result, and result from combination of FGPGM and optimization are compared.

sampling cycle. The performances of these three methods will be discussed in §3.4. The last step is to solve the following minimization problem:

$$\min_{\boldsymbol{\theta}} \|\boldsymbol{x}_M(\boldsymbol{\theta}) - \boldsymbol{y}\|^2_{L^2(0,T)} = \min_{\boldsymbol{\theta}} \sum_{i=1}^{N_2} |\boldsymbol{x}_M(t_i) - \boldsymbol{y}(t_i)|^2. \tag{2.11}$$

In the optimization process, gradient descent method is adopted where numerical gradient is used in each searching step. One advantage of least-square optimization is its ability to obtain accurate results at low computational cost, especially for observations with Gaussian noise. But it requires proper initial guess of the parameters so as to avoid falling in local minima, whereas gradient matching is relatively less sensitive to the initial guess. However, for FGPGM a large amount of MCMC samplings are necessary to ensure the expectations of random variables make sense, which increases computational cost. Therefore, if we combine these two methods, it is possible to use just a few MCMC samplings to obtain a good initial guess for the least-square optimization.

# 3. Experiments

For the Gaussian regression step for observable variables, the code published alongside Wenk *et al.* [14] was used. The gradient matching part should then be adapted to the partial observation case according to figure 1. In the following, we refer to FGPGM as the modified fast Gaussian process gradient matching algorithm that is adapted for partial observation case.

## 3.1. Lotka Volterra

The Lotka Volterra system was originally proposed in Lotka [16]. It was introduced to model the prey–predator interaction system whose dynamics is given by

$$\dot{x}_1 = \theta_1 x_1(t) - \theta_2 x_1(t) x_2(t) \tag{3.1}$$

and

$$\dot{x}_2 = -\theta_3 x_2(t) + \theta_4 x_1(t) x_2(t), \tag{3.2}$$

where $\theta_1, \theta_2, \theta_3, \theta_4 > 0$. In the present work, the system was observed with one variable and the initial value of the other variable. The other set-up is the same as Gorbach *et al.* [11] and Wenk *et al.* [14]. The observed series are located in the time interval [0, 2] at 20 uniformly distributed observation times. The initial values of the variables are (5, 3). The history of the observable variable is generated with numerical integration of the system with true parameters $\boldsymbol{\theta}^* = (\theta_1, \theta_2, \theta_3, \theta_4) = (2, 1, 4, 1)$ added with Gaussian noise with standard deviation 0.1. The radial basis function kernel (RBF) was used for the Gaussian process. For the model uncertainty, we set $\gamma = 0.3$. For the FGPGM step, the burn-in number and valid number are $N_{\text{burnin}} = 7500$, $N_{\text{MCMC}} = 10\,000$, respectively. The results with $x_1$ being observed are shown in figure 2. Those with observation of $x_2$ are given in figure 3. The errors of the reconstructed solution and parameters are summarized in tables 1–4. At this noise level, the inferences with and without optimization step both perform well with the overall relative error being of order $10^{-2}$. In the case with measurement of $x_2$, we can see that the optimization process can improve the results from FGPGM, with the discernable errors of $\theta_1$ and $\theta_3$ in figure 3c being reduced. Although $\theta_2$

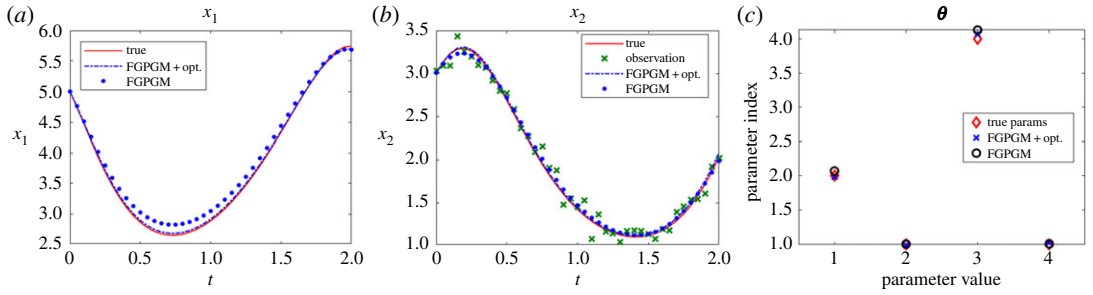

**Figure 3.** The state evolution over time for Lotka–Volterra system and parameter inference results. $x_2$ is observable and $x_1$ is latent variable. The ground truth, FGPGM result, and result from combination of FGPGM and optimization are compared. (a) $x_1$, (b) $x_2$, (c) $\boldsymbol{\theta}$.

**Table 1.** Error of reconstructed solution with $x_1$ being observable.

|  | $\frac{\|x_1-x_1^*\|_{l^\infty}}{\|x_1^*\|_{l^\infty}}$ | $\frac{\|x_1-x_1^*\|_{l^2}}{\|x_1^*\|_{l^2}}$ | $\frac{\|x_2-x_2^*\|_{l^\infty}}{\|x_2^*\|_{l^\infty}}$ | $\frac{\|x_2-x_2^*\|_{l^2}}{\|x_2^*\|_{l^2}}$ |
|---|---|---|---|---|
| FGPGM | $1.69 \times 10^{-2}$ | $0.83 \times 10^{-2}$ | $3.47 \times 10^{-2}$ | $2.44 \times 10^{-2}$ |
| FGPGM + Opt. | $1.40 \times 10^{-2}$ | $0.49 \times 10^{-2}$ | $4.26 \times 10^{-2}$ | $3.63 \times 10^{-2}$ |

**Table 2.** Error of reconstructed parameters with $x_1$ being observable.

|  | $|\theta_1 - \theta_1^*|$ | $|\theta_2 - \theta_2^*|$ | $|\theta_3 - \theta_3^*|$ | $|\theta_4 - \theta_4^*|$ |
|---|---|---|---|---|
| FGPGM | $8.00 \times 10^{-2}$ | $4.79 \times 10^{-2}$ | $8.32 \times 10^{-2}$ | $2.35 \times 10^{-2}$ |
| FGPGM + Opt. | $9.82 \times 10^{-2}$ | $1.88 \times 10^{-2}$ | $8.39 \times 10^{-2}$ | $1.17 \times 10^{-2}$ |

**Table 3.** Error of reconstructed solution with $x_2$ being observable.

|  | $\frac{\|x_1-x_1^*\|_{l^\infty}}{\|x_1^*\|_{l^\infty}}$ | $\frac{\|x_1-x_1^*\|_{l^2}}{\|x_1^*\|_{l^2}}$ | $\frac{\|x_2-x_2^*\|_{l^\infty}}{\|x_2^*\|_{l^\infty}}$ | $\frac{\|x_2-x_2^*\|_{l^2}}{\|x_2^*\|_{l^2}}$ |
|---|---|---|---|---|
| FGPGM | $2.99 \times 10^{-2}$ | $3.08 \times 10^{-2}$ | $1.98 \times 10^{-2}$ | $1.92 \times 10^{-2}$ |
| FGPGM + Opt. | $0.80 \times 10^{-2}$ | $0.65 \times 10^{-2}$ | $1.37 \times 10^{-2}$ | $1.07 \times 10^{-2}$ |

**Table 4.** Error of reconstructed parameters with $x_2$ being observable.

|  | $|\theta_1 - \theta_1^*|$ | $|\theta_2 - \theta_2^*|$ | $|\theta_3 - \theta_3^*|$ | $|\theta_4 - \theta_4^*|$ |
|---|---|---|---|---|
| FGPGM | $7.16 \times 10^{-2}$ | $0.06 \times 10^{-2}$ | $1.34 \times 10^{-1}$ | $0.07 \times 10^{-2}$ |
| FGPGM + Opt. | $1.79 \times 10^{-2}$ | $1.76 \times 10^{-2}$ | $7.77 \times 10^{-2}$ | $1.88 \times 10^{-2}$ |

and $\theta_4$ appear quite accurate in the case of using FGPGM only ($N_{\text{burnin}} = 7500$, $N_{\text{MCMC}} = 10\,000$) in table 4, the relatively large error in $\theta_3$ may affect the reconstructed solution due to the coupling effect of the parameters in the system. By contrast, the parameter errors after the combination of FGPGM and optimization are all of order $10^{-2}$.

The sensitivities of the variables to the parameters are listed in table 5.

The sensitivity indexes at the true parameter set $\boldsymbol{\theta}^*$ are defined as

$$S_{ij} = \frac{1}{\|x_i\|_{L_2(T_1,T_2)}} \left\| \frac{\partial x_i(t; \boldsymbol{\theta})}{\partial \theta_j} \right\|_{L_2(T_1,T_2)} (\boldsymbol{\theta}^*) \tag{3.3}$$

which are normalized. It is approximated by numerical difference. It is shown that near the true parameter set, $\theta_1$ and $\theta_3$ are relatively less sensitive to the variables than other parameters. This explains that $\theta_1$ and $\theta_3$ are less accurate in the numerical test (figures 2c and 3c).

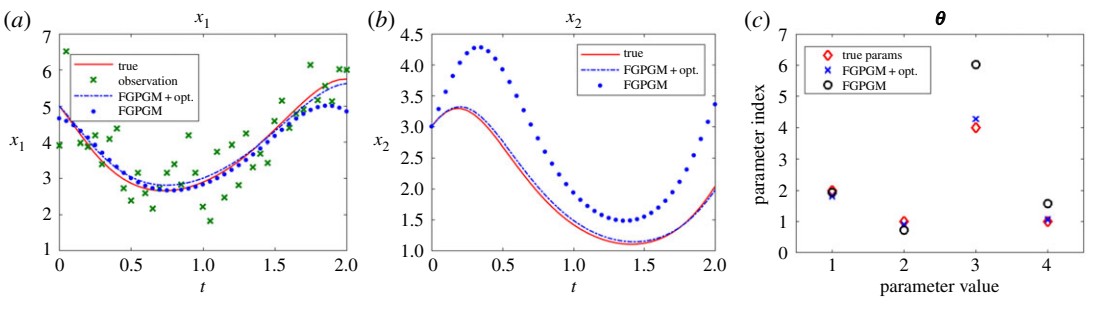

**Figure 4.** Reconstruction and inference results for the Lotka–Volterra system with $x_1$ being observable and $x_2$ latent. The ground truth, FGPGM result, and result from combination of FGPGM and optimization are compared. The observation noise has a standard deviation 0.5 (large noise case). (a) $x_1$, (b) $x_2$, (c) $\boldsymbol{\theta}$.

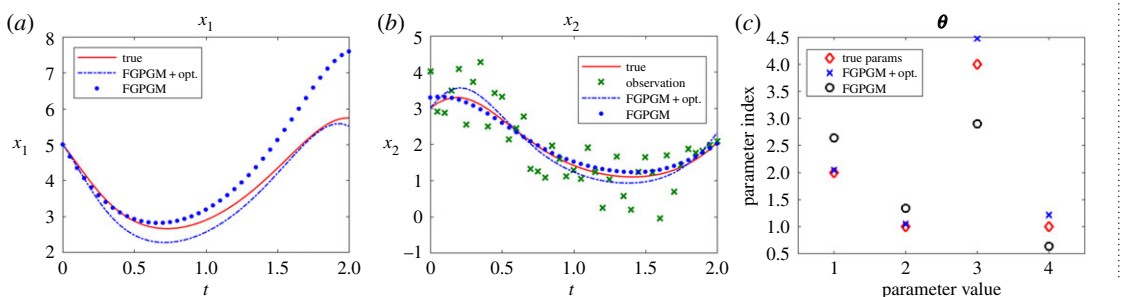

**Figure 5.** The state evolution over time for Lotka–Volterra system. $x_2$ is observable and $x_1$ is latent variable. The ground truth, FGPGM result, and result from combination of FGPGM and optimization are compared. The observation noise has a standard deviation 0.5 (large noise case). (a) $x_1$, (b) $x_2$, (c) $\boldsymbol{\theta}$.

**Table 5.** Sensitivity of each variable to parameters for Lotka–Volterra system at $\boldsymbol{\theta}^* = (2, 1, 4, 1)$. The sensitivity index is defined in equation (3.3).

| $S_{ij}$ | $x_1$ | $x_2$ |
|---|---|---|
| $\theta_1$ | 0.20 | 0.61 |
| $\theta_2$ | 0.52 | 1.13 |
| $\theta_3$ | 0.40 | 0.33 |
| $\theta_4$ | 1.27 | 0.98 |

**Table 6.** Error of reconstructed solution with $x_1$ being observable (large noise).

| | $\dfrac{\|x_1 - x_1^*\|_{l^\infty}}{\|x_1^*\|_{l^\infty}}$ | $\dfrac{\|x_1 - x_1^*\|_{l^2}}{\|x_1^*\|_{l^2}}$ | $\dfrac{\|x_2 - x_2^*\|_{l^\infty}}{\|x_2^*\|_{l^\infty}}$ | $\dfrac{\|x_2 - x_2^*\|_{l^2}}{\|x_2^*\|_{l^2}}$ |
|---|---|---|---|---|
| FGPGM | $1.56 \times 10^{-1}$ | $7.30 \times 10^{-2}$ | $4.03 \times 10^{-1}$ | $3.81 \times 10^{-1}$ |
| FGPGM+Opt. | $3.02 \times 10^{-2}$ | $3.08 \times 10^{-2}$ | $2.70 \times 10^{-2}$ | $2.68 \times 10^{-2}$ |

**Table 7.** Error of reconstructed parameters with $x_1$ being observable (large noise).

| | $|\theta_1 - \theta_1^*|$ | $|\theta_2 - \theta_2^*|$ | $|\theta_3 - \theta_3^*|$ | $|\theta_4 - \theta_4^*|$ |
|---|---|---|---|---|
| FGPGM | $6.37 \times 10^{-2}$ | $2.76 \times 10^{-1}$ | 2.01 | $5.75 \times 10^{-1}$ |
| FGPGM+Opt. | $1.78 \times 10^{-1}$ | $1.06 \times 10^{-1}$ | $2.75 \times 10^{-1}$ | $5.51 \times 10^{-2}$ |

The cases with larger measurement noise level (with standard deviation 0.5) are shown in figures 4 and 5, corresponding to $x_1$ and $x_2$ observations, respectively. The errors of reconstructed solution and identified parameters are summarized in tables 6–9. It can be seen that the prediction of the unknown variable can deviate far from the ground truth if we use FGPGM method only (without Step 3 in the

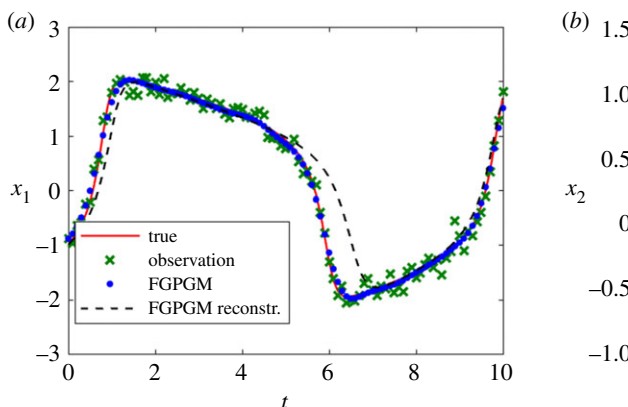 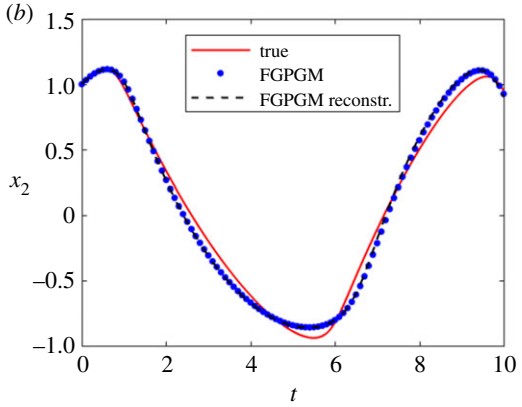

**Figure 6.** Results for FHN system obtained from FGPGM method, without further optimization. $x_1$ is observable and $x_2$ is latent variable. The ground truth, FGPGM result and reconstructed solution (integration of ODEs with inferred parameters) are compared. (a) $x_1$, (b) $X_2$.

**Table 8.** Error of reconstructed solution with $x_2$ being observable (large noise).

| | $\frac{\|x_1 - x_1^*\|_{L^\infty}}{\|x_1^*\|_{L^\infty}}$ | $\frac{\|x_1 - x_1^*\|_{L^2}}{\|x_1^*\|_{L^2}}$ | $\frac{\|x_2 - x_2^*\|_{L^\infty}}{\|x_2^*\|_{L^\infty}}$ | $\frac{\|x_2 - x_2^*\|_{L^2}}{\|x_2^*\|_{L^2}}$ |
|---|---|---|---|---|
| FGPGM | $3.22 \times 10^{-1}$ | $1.89 \times 10^{-1}$ | $8.80 \times 10^{-2}$ | $5.34 \times 10^{-2}$ |
| FGPGM + Opt. | $6.88 \times 10^{-2}$ | $6.68 \times 10^{-2}$ | $8.40 \times 10^{-2}$ | $7.71 \times 10^{-2}$ |

**Table 9.** Error of reconstructed parameters with $x_2$ being observable (large noise).

| | $|\theta_1 - \theta_1^*|$ | $|\theta_2 - \theta_2^*|$ | $|\theta_3 - \theta_3^*|$ | $|\theta_4 - \theta_4^*|$ |
|---|---|---|---|---|
| FGPGM | $6.40 \times 10^{-1}$ | $3.44 \times 10^{-1}$ | $1.10$ | $3.63 \times 10^{-1}$ |
| FGPGM + Opt. | $4.35 \times 10^{-2}$ | $4.35 \times 10^{-2}$ | $4.72 \times 10^{-1}$ | $2.16 \times 10^{-1}$ |

algorithm above). The inferring of states and parameters can be improved after further applying the deterministic optimization. This improvement is more obvious than the low noise case.

## 3.2. Spiky dynamics

This example is a system proposed by FitzHugh [17] and Nagumo *et al.* [18] for modelling the spike potentials in the giant squid neurons, which is abbreviated as FHN system. This system involves two ODEs (3.4) and (3.5) with three parameters.

$$\dot{x}_1 = \theta_1 \left( x_1 - \frac{x_1^3}{3} + x_2 \right) \tag{3.4}$$

and

$$\dot{x}_2 = -\frac{1}{\theta_1}(x_1 - \theta_2 + \theta_3 x_2). \tag{3.5}$$

The FHN system has notoriously fast changing dynamics due to its highly nonlinear terms. In the following numerical tests, the Matern 5/2 [19] kernel was used and $\gamma$ was set to 0.3, the same as that in Wenk *et al.* [14]. We assume one of the two variables is observable, which was generated with $\boldsymbol{\theta}^* = (\theta_1, \theta_2, \theta_3) = (0.2, 0.2, 3)$ and added by Gaussian noise with average signal-to-noise ratio SNR = 100. There were 100 data points uniformly spaced in [0, 10]. The burn-in number and valid number of samplings in the FGPGM step are $N_{\text{burnin}} = 7500$, $N_{\text{MCMC}} = 10\,000$, respectively.

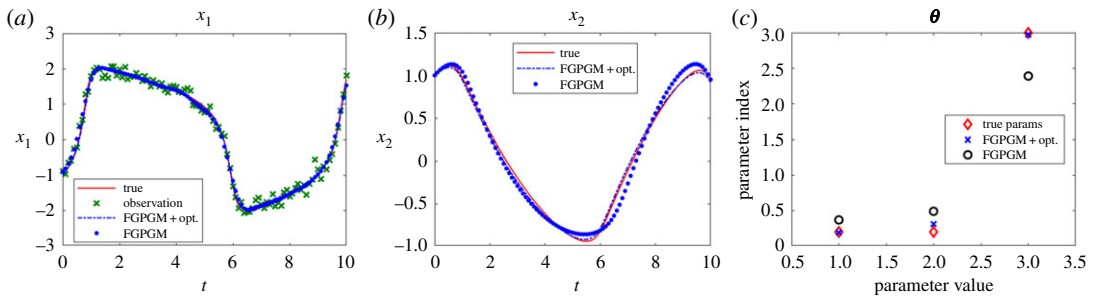

**Figure 7.** The state evolution over time and identified parameters for FHN system. $x_1$ is observable and $x_2$ is latent variable. The ground truth, FGPGM result, and result from combination of FGPGM and optimization are compared. (a) $x_1$, (b) $x_2$, (c) $\boldsymbol{\theta}$.

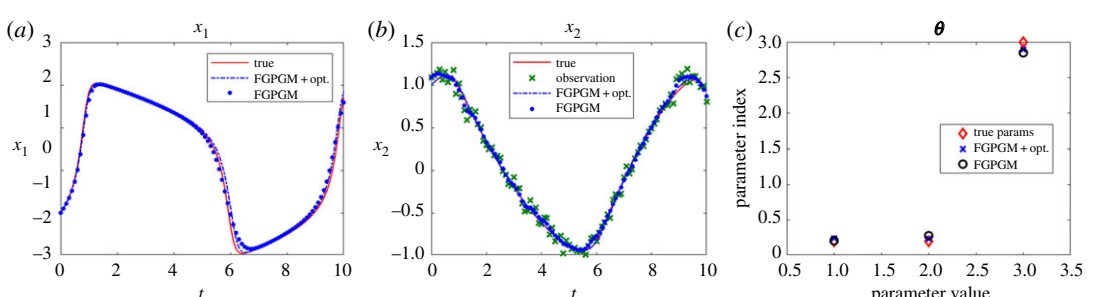

**Figure 8.** The state evolution over time and identified parameters for FHN system. $x_2$ is observable and $x_1$ is latent variable. The ground truth, FGPGM result, and result from combination of FGPGM and optimization are compared. (a) $x_1$, (b) $x_2$, (c) $\boldsymbol{\theta}$.

**Table 10.** Sensitivity of each variable to parameters for FHN system at $\boldsymbol{\theta}^* = (0.2, 0.2, 3.0)$. The sensitivity index is defined as equation (3.3).

| $S_{ij}$ | $x_1$ | $x_2$ |
|---|---|---|
| $\theta_1$ | 2.33 | 1.24 |
| $\theta_2$ | 0.44 | 0.31 |
| $\theta_3$ | 1.01 | 0.55 |

In this case, if we merely use FGPGM step, the reconstructed solution corresponding to the identified parameters may deviate significantly from the true time series (see figure 6, where data of $x_1$ are observable). It was pointed out [14] that all GP-based gradient matching algorithms lead to smoother trajectories than the ground truth. This becomes more severe with sparse observation. The least-square optimization after FGPGM may well reduce this effect and give a better reconstruction of the solution (figure 7).

Figures 7 and 8 present the results with single $x_1$ and $x_2$ observations, respectively. In both cases, the identified parameters are more accurate than using FGPGM only. From the sensitivity check in table 10, it is expected that $\theta_1$ is most accurate because it is most sensitive among these three parameters, whereas $\theta_2$ is most insensitive and would be harder to be identified. The numerical results agree with that. It is worth mentioning that in the FGPGM step, only 3500 samplings were taken and the time for optimization step was much less than FGPGM step. This means the time needed for the whole process can be greatly saved compared with that in Wenk *et al.* [14], where 100 000 MCMC samplings were implemented.

In this example, we also notice that if we merely use least-square optimization method, the local minimum effect would lead to reconstruction being far from the ground truth, which is even less robust than FGPGM method. For example, if we choose initial guess of the parameters near $(\theta_1, \theta_2, \theta_3) = (1.51, 2.2, 1.78)$ then the cost functional will fall into the local minimum during gradient-based search (figure 9). The existence of many local minima in the full observation case has been pointed out in e.g. [7,20]. These results clearly illustrate the performance of the combination of FGPGM and least-square optimization to avoid local minimum solution.

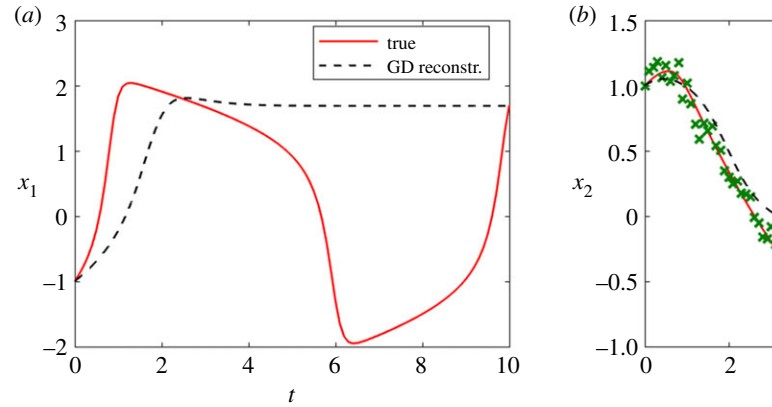

**Figure 9.** Results of FHN system with $x_1$ being latent, obtained by merely using least-square optimization with initial guess of parameters near a local minimum point. (a) $x_1$, (b) $x_2$.

## 3.3. Protein transduction

Finally, the protein transduction system proposed in Vyshemisky & Girolami [21] was adopted to illustrate the performance of the method in ODEs with more equations. The system is described by

$$\dot{x}_1 = -\theta_1 x_1 - \theta_2 x_1 x_3 + \theta_3 x_4, \tag{3.6}$$

$$\dot{x}_2 = \theta_1 x_1, \tag{3.7}$$

$$\dot{x}_3 = -\theta_2 x_1 x_3 + \theta_3 x_4 + \theta_5 \frac{x_5}{\theta_6 + x_5}, \tag{3.8}$$

$$\dot{x}_4 = \theta_2 x_1 x_3 - \theta_3 x_4 - \theta_4 x_4 \tag{3.9}$$

and

$$\dot{x}_5 = \theta_4 x_4 - \theta_5 \frac{x_5}{\theta_6 + x_5}. \tag{3.10}$$

We adopted the same experimental set-up of Dondelinger *et al.* [10] and Wenk *et al.* [14] as follows. $\gamma = 10^{-4}$ in FGPGM step. The observation were made at discrete times [0, 1, 2, 4, 5, 7, 10, 15, 20, 30, 40, 50, 60, 80, 100]. The initial condition was [1, 0, 1, 0, 0] and the data were generated by numerically integrating the system under $\boldsymbol{\theta}^* = [0.07, 0.6, 0.05, 0.3, 0.017, 0.3]$, added by Gaussian noise with standard deviation 0.01. A sigmoid kernel was used to deal with the logarithmically spaced observation times and the typically spiky form of the dynamics as in the previous papers.

Figure 10 gives the result with $x_3$ being unobserved. In fact, the situations with one of other variables being unknown have better results than the case illustrated here, which will not be presented here. We can see that $x_3$ was not well fitted by merely using FGPGM step ($N_{\text{burnin}} = 7500$, $N_{\text{MCMC}} = 10\,000$), whereas further applying the least-square optimization generates satisfactory results, with the parameters $\theta_2$ and $\theta_4$ being significantly improved. The sensitivity check is summarized in table 11, from which we can see that $\theta_2$ is less sensitive and thereby harder to infer accurately.

It was mentioned in Dondelinger *et al.* [10] that $\theta_5$ and $\theta_6$ in this system are difficult to fit. $\theta_5/\theta_6$ is relatively sensitive but is still likely to be overestimated. It was pointed out in [14] that the parameters of the protein transduction system are unidentifiable. In the present case with partial observation, $\theta_6$ is far beyond the truth with the inferred value being around 3.0, which is therefore not presented in the figure. However, the observation can still be acceptably reconstructed.

It would also be of interest to see the performance of our method for the cases with more latent variables. In this model, although $x_2$ is not involved in equations for other variables, the data of $x_2$ helps infer $\theta_1$. We also notice that $\dot{x}_3 + \dot{x}_4 + \dot{x}_5 = 0$, which means with known initial conditions the histories of the three variables can be known from any two of them. If $x_1$ and $x_2$ are both missing, it is impossible to identify $\theta_1$. Therefore, in the following test we choose data of $x_2$, $x_4$ and $x_5$ as observations. The data have a Gaussian noise with standard deviation 0.01, the same as the previous case with one latent variable. It can be seen from figure 11 that the result from FGPGM step is worse than the case with only one latent variable, but the final reconstruction of latent variables and parameter identification after Step 3 is not significantly different from the case with one latent variable.

In order to imitate real situations, we considered the case with abnormal measurements, in which some observations have significant deviation from the ground truth. Moreover, only $x_1$ and $x_5$ are observable. Note that further reducing the observable variable will make the system unidentifiable.

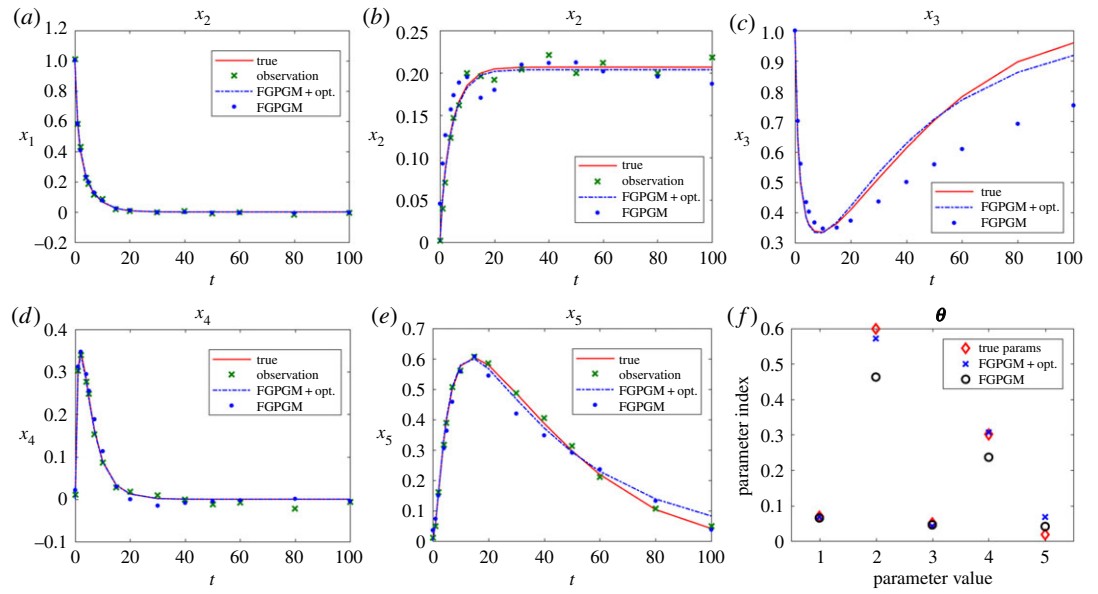

**Figure 10.** The state evolution over time and inferred parameters for protein transduction system. $x_3$ is unknown and other variables are observable. The ground truth, FGPGM result, and result from combination of FGPGM and optimization are compared. (a) $x_1$, (b) $x_2$, (c) $x_3$, (d) $x_4$, (e) $x_5$, (f) $\boldsymbol{\theta}$.

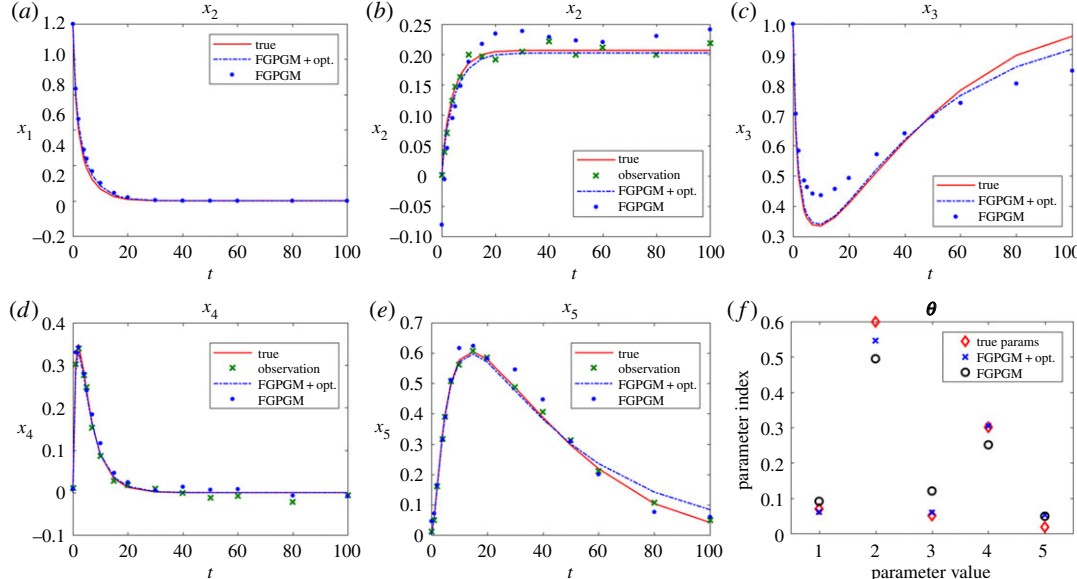

**Figure 11.** The state evolution over time and inferred parameters for protein transduction system. $x_1$ and $x_3$ are unknown and other variables are observable. The ground truth, FGPGM result, and result from combination of FGPGM and optimization are compared. (a) $x_1$, (b) $x_2$, (c) $x_3$, (d) $x_4$, (e) $x_5$, (f) $\boldsymbol{\theta}$.

**Table 11.** Sensitivity of each variable to parameters for protein transduction system at $\boldsymbol{\theta}^* = [0.07, 0.6, 0.05, 0.3, 0.017, 0.3]$. The sensitivity index is defined as equation (3.3).

| $S_{ij}$ | $x_1$ | $x_2$ | $x_3$ | $x_4$ | $x_5$ |
|---|---|---|---|---|---|
| $\theta_1$ | 2.86 | 9.78 | 1.73 | 1.77 | 3.33 |
| $\theta_2$ | 0.70 | 0.98 | 0.22 | 0.59 | 0.41 |
| $\theta_3$ | 1.35 | 2.11 | 0.47 | 0.92 | 0.90 |
| $\theta_4$ | 0.26 | 0.43 | 0.03 | 2.64 | 0.62 |
| $\theta_5$ | 1.53 | 2.58 | 24.48 | 0.90 | 49.38 |
| $\theta_6$ | 0.04 | 0.07 | 0.60 | 0.02 | 1.21 |

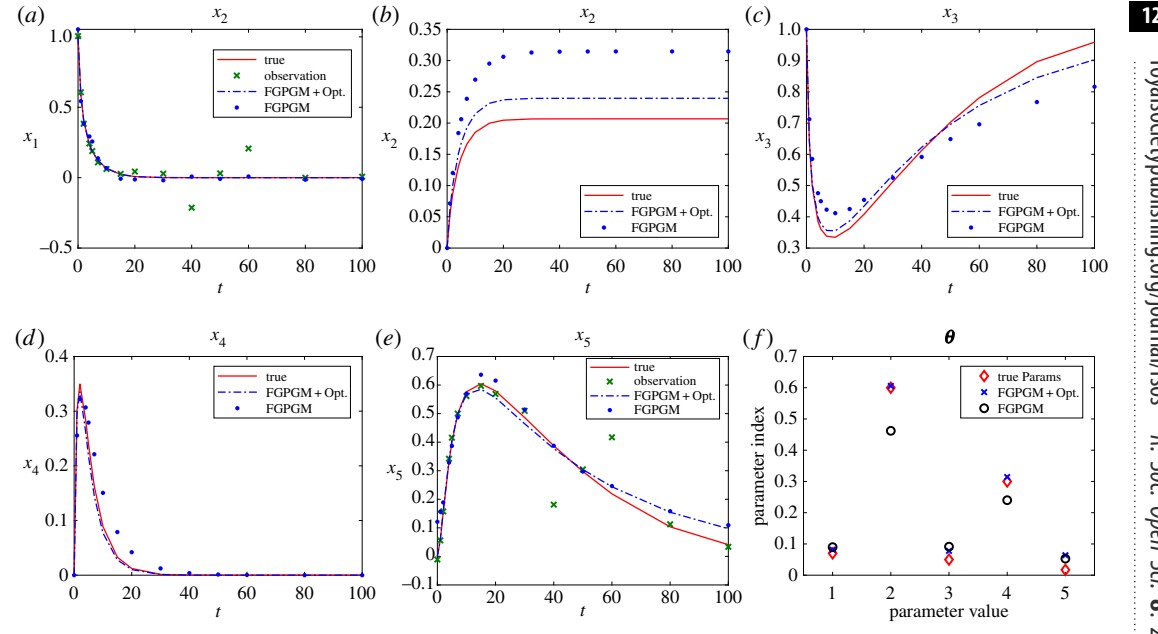

**Figure 12.** The state evolution over time and inferred parameters for protein transduction system. $x_2$, $x_3$, $x_4$ are unknown and other variables are observable. The ground truth, FGPGM result, and result from combination of FGPGM and optimization are compared. Observations at $t = 40$, $60$ are abnormal with significant deviation from ground truth. (a) $x_1$, (b) $x_2$, (c) $x_3$, (d) $x_4$, (e) $x_5$, (f) $\boldsymbol{\theta}$.

**Table 12.** Error of inferred parameters under various noise levels.

| noise level (s.d.) | method | $|\theta_1 - \theta_1^*|$ | $|\theta_2 - \theta_2^*|$ | $|\theta_3 - \theta_3^*|$ | $|\theta_4 - \theta_4^*|$ | $|\theta_5 - \theta_5^*|$ |
|---|---|---|---|---|---|---|
| 0.005 | FGPGM | 0.0127 | 0.1009 | 0.0007 | 0.0312 | 0.0372 |
| 0.01 | FGPGM | 0.0055 | 0.1312 | 0.0590 | 0.1316 | 0.0308 |
| 0.02 | FGPGM | 0.0193 | 0.1730 | 1.1257 | 1.4630 | 0.0272 |
| 0.005 | FGPGM + Opt. | 0.0253 | 0.0489 | 0.0327 | 0.0638 | 0.0602 |
| 0.01 | FGPGM + Opt. | 0.0267 | 0.0702 | 0.0270 | 0.0736 | 0.0704 |
| 0.02 | FGPGM + Opt. | 0.0121 | 0.1063 | 1.0824 | 1.4883 | 0.0477 |

The results are shown in figure 12. At $t = 40$ and $t = 60$, the measurement errors are abnormally large (figure 12$a$,$e$). The identified parameters and reconstructed state series seem not severely impacted by the large measurement errors, which shows some robustness. This also illustrates that the sigmoid kernel is a proper choice for this example. The performance in more real situations is worth further investigation, especially for complex dynamical systems.

Besides, the dependence of errors of inferred parameter on the noise levels have also been carried out. Table 12 gives the comparison among observation noise levels. The measurement errors are Gaussian and independent and identically distributed. Here $x_1$ and $x_5$ are observable. On the whole the identified parameters become less accurate when the observation noise level increases. When the standard deviation increases to 0.02, some parameters show quite large errors such that the identification should be regarded as failed. We need to mention that small errors in part of the parameters does not necessarily imply a better reconstruction of solution or prediction. It relates to the sensitivities of parameters and variables, which need further investigation.

## 3.4. Comparisons among treatments for unobservable variables

Now we illustrate the performances of the three ways to infer the unobservable variables proposed in §2. We take the spiky dynamics in §3.2 as example and choose $x_2$ as unobservable variable. Here the observation noise for $x_1$ is increased with a standard deviation of 0.1 and the time resolution is reduced to 50 points. The results of these three methods are compared in figures 13–17. In all the cases, 2000 burning cycles and 3000 sampling cycles are implemented. In the full integration case, the ode45 solver is applied, while in the partial

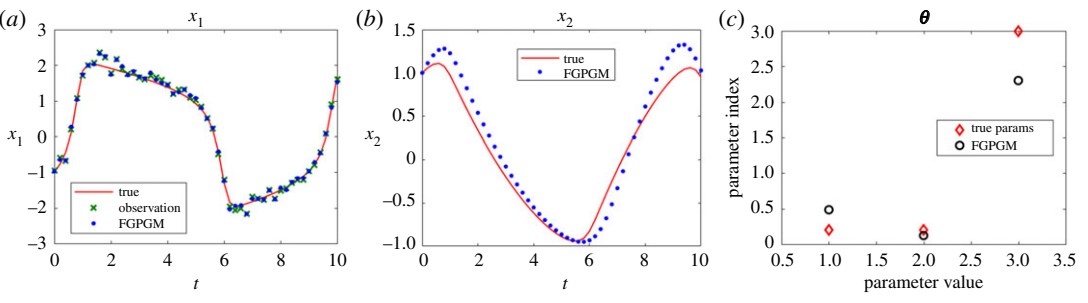

**Figure 13.** The state evolution over time and identified parameters for FHN system. $x_2$ is unobservable and inferred by the integration of the whole ODE system. (*a*) $x_1$, (*b*) $x_2$, (*c*) $\boldsymbol{\theta}$.

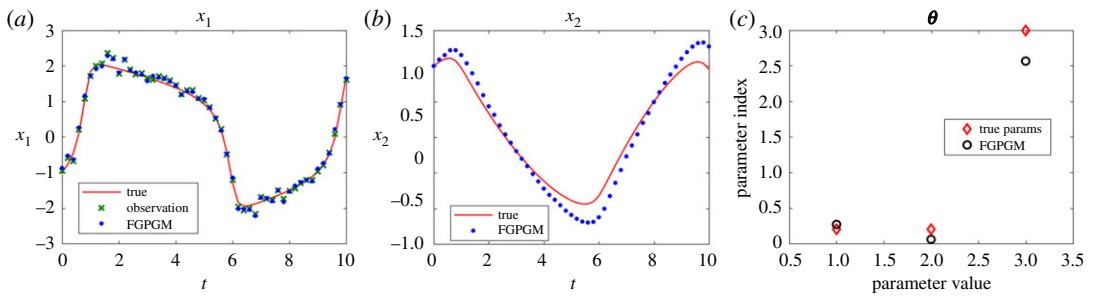

**Figure 14.** The state evolution over time and identified parameters for FHN system. $x_2$ is unobservable and inferred by integrating the equation for $x_2$ with inferred $x_1$ as known coefficient. (*a*) $x_1$, (*b*) $x_2$, (*c*) $\boldsymbol{\theta}$.

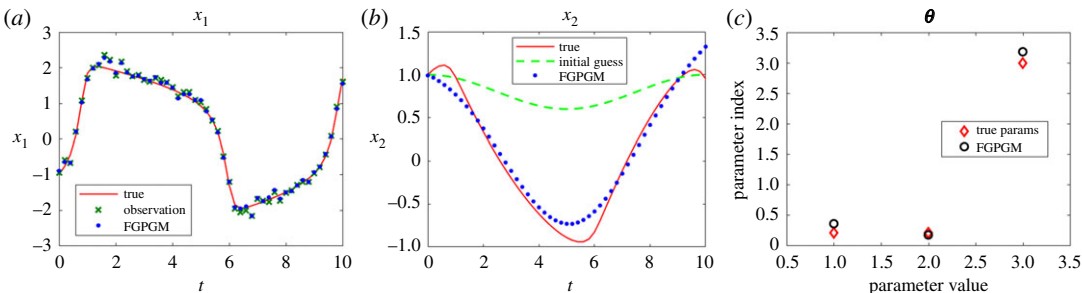

**Figure 15.** The state evolution over time and identified parameters for FHN system. $x_2$ is unobservable and inferred by sampling under the assumption of Gaussian process. (*a*) $x_1$, (*b*) $x_2$, (*c*) $\boldsymbol{\theta}$.

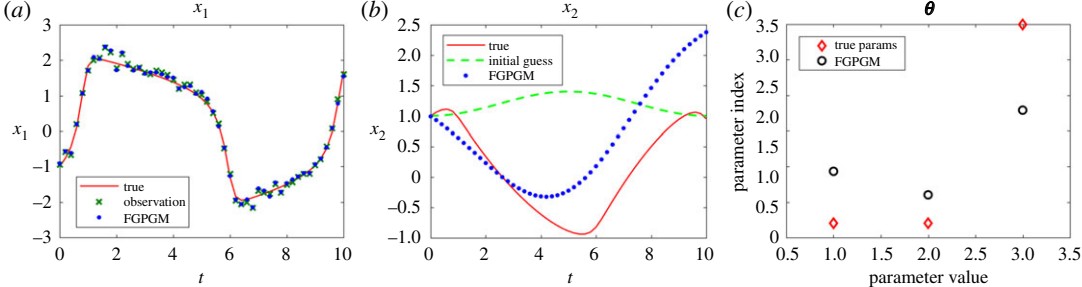

**Figure 16.** The state evolution over time and identified parameters for FHN system. $x_2$ is unobservable and inferred by sampling under the assumption of Gaussian process. The initial guess of $x_2$ is different from that in figure 15*b*. (*a*) $x_1$, (*b*) $x_2$, (*c*) $\boldsymbol{\theta}$.

integration case Eular formula is adopted. Other computational parameters are unified as $\gamma = 1.0 \times 10^{-3}$, standard deviations that control the sampling steps are 0.001 and 0.12 for $x_1$ and $\boldsymbol{\theta}$, respectively. The initial guess for the unobservable variable are plotted in dashed line in figure 15*b*.

The total CPU times corresponding to the full integration, partial integration and sampling approaches are 207.6 s, 180.7 s and 529.4 s correspondingly. In the partial integration case, if denoise and interpolation for observed variables or high-order integration scheme are needed to ensure the stability and convergence, then the computational cost will increase. If most of the variables in a high-dimensional system are

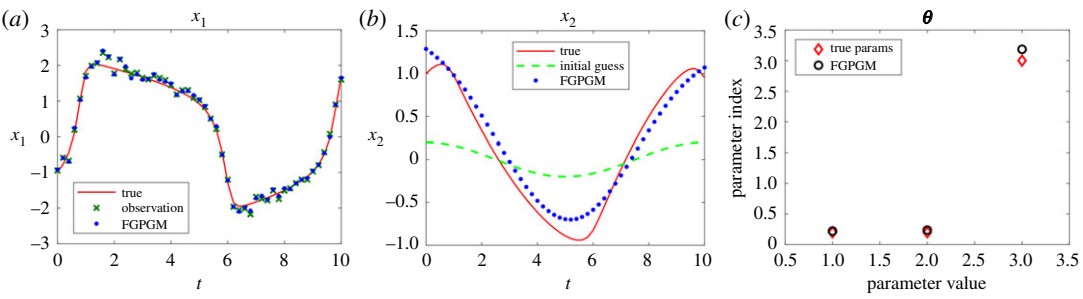

**Figure 17.** The state evolution over time and identified parameters for FHN system. $x_2$ is unobservable and inferred by sampling under the assumption of Gaussian process. The initial guess of $x_2$ at $t = 0$ is also far from the ground truth. (a) $x_1$, (b) $x_2$, (c) $\boldsymbol{\theta}$.

unobservable, then the partial integration may not have a significant time saving compared with the full integration case. At present, the system is so simple that the computational expense for integration is very low compared with sampling and probability calculation. Therefore, the CPU time is much higher in the third approach where the sampled states are doubled. However, when the system is complex such that the computational cost for solving the system becomes an issue, the sampling approach with proper initial guesses may show advantages since it depends on the time points, and sparse approximation works well in Gaussian process methods [8]. It can be seen from the results that the sampling approach gets a better parameter approximation than the integration methods (figure 15c), though the inference of $x_2$ is worse than them in terms of the curve features. It should be noted that the results for the full sampling method depend on the initial guess. An inappropriate initial distribution may lead to large deviation from the truth at the same computational parameters (figure 16b,c). If the system is large and most of the variables are unobservable, then the initial guesses of them may become an issue. To investigate the case of unknown initial conditions, we propose an initial guess such that $x_2$ at $t = 0$ deviates a lot from the ground truth (figure 17b). It can be seen that the sampling is still able to find a solution close to the truth for this system. We should point out that lack of initial conditions may lead to non-uniqueness of identification for some systems [22].

According to the above discussions, we suggest that if the system is very small such that solving ODEs is cheap in time, one can adopt the integration approaches since they are independent of the prior information of the unobserved variables. If the integration requires fine time step and most of the variables in the system are unobservable, then the full integration is preferred since denoise and interpolation of the observable variables which appear in the equations for unobservable ones can be avoided. For large-scale problems with reliable prior information, the full sampling method may have advantages in both accuracy and efficiency.

## 3.5. Indirect observation

In this section, we consider the case that the observable output $y(t)$ is a function of the original system variable $x(t)$. Then, the observation with noise becomes $Y(t) = y(x(t)) + \varepsilon$. Take the following system as example:

$$\dot{x}_1 = \theta_1 x_2\, e^{-\theta_2 x_1} \tag{3.11}$$

and

$$\dot{x}_2 = x_1 \tag{3.12}$$

and the measurement equation is

$$y = e^{x_1}. \tag{3.13}$$

For such problem, we take time derivative of the output function

$$\dot{y} = \theta_1 x_2\, e^{(1-\theta_2)x_1},$$

and then have a new three-equation system for $\{x_1, x_2, y\}$, in which the first two variables are unobservable and the last one is observable. The above method can then be applied to the new system. Assume that the observation is taken at 25 uniformly distributed time points within $[0, 5]$, with a Gaussian noise of standard deviation 0.2. The true values of the parameters are $\boldsymbol{\theta}^* = (0.5, 1)$, and initial values $x(0) = [0.5, 0]$. In the Gaussian process gradient matching step, the Matern 5/2 kernel was used and $\gamma = 10^{-3}$. The numerical results are provided in figure 18. The burning number and total sampling number are 2000 and 5000,

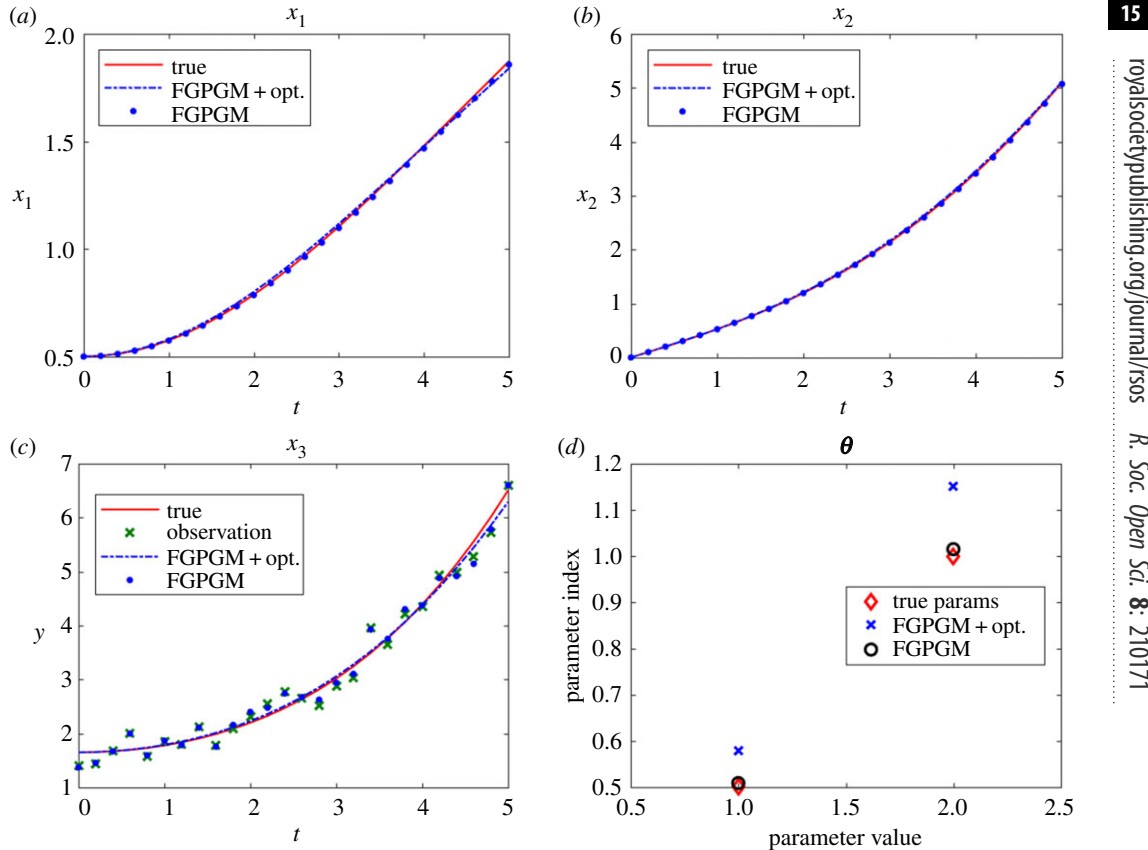

**Figure 18.** The state evolution over time and identified parameters for example 4. Only $y(t) = e^{x_1(t)}$ is observable. (a) $x_1$, (b) $x_2$, (c) $y$, (d) $\boldsymbol{\theta}$.

respectively. In this example, the FGPGM step has already reached a satisfactory inference for the parameters (figure 18$d$), and further application of least optimization loses accuracy at the end of the time interval due to sparsity of the data (figure 18$c$) which leads to deviations in the parameters.

# 4. Discussion

In the work, we proposed an effective method for parameter inference of coupled ODE systems with partially observable data. Our method is based on previous work known as FGPGM [14], which avoids product of experts heuristics. We provide the treatment of latent variable and graphical frame for such problems. The likelihood formula is then derived. In order to improve the accuracy and efficiency of the method, we also use least-square optimization in our computation. In our numerical tests, we use only 10% of the sampling number that is suggested in the literature for the FGPGM step, which can already provide a good initial guess for the least optimization. Owing to the existence of latent variables, three approaches for treatment of unobservable variables are provided with the performances discussed. Suggestions for choices of the approaches are given according to the system scales and reliability of prior information. The present algorithm is robust and accurate with large data noise and partial observations. At the same time, thanks to the step of Gaussian process-based inference, it is easier to get rid of local minima for the least-square optimization step and achieve the global minimum, which reduces the dependence on the initial guess of parameters.

Data accessibility. Data and relevant code for this research work are stored in GitHub: https://github.com/yuchensufe/ PartialObservation-GPGM/tree/v1.0 and have been archived within the Zenodo repository: https://doi.org/10.5281/ zenodo.4501573.

Authors' contributions. Y.C. and S.X. designed the algorithm. Y.C. conducted numerical simulations. Y.C., J.C., A.G., H.H. and S.X. contributed to the research conceptualization and prepared the manuscript.

Competing interests. We declare we have no competing interests.

Funding. This work was supported in part by Kunshan Government Research Fund (grant no. R97030012S 2020DKU0032 (S1310)), Nature Science and Research Council (NSERC) of Canada and the Fields Institute for Research in Mathematical Sciences.

**Acknowledgements.** The authors also express gratitude to Prof. Xin Gao, Nathan Gold and other members of the Fields CQAM Lab on Health Analytics and Modelling for very beneficial discussions.

# Appendix A. Preliminaries

In the following, we list some preliminaries on derivatives of a Gaussian process that are used in this work, the proofs can be find in e.g. [14,19,23]. Denote a random process $X_t$, its realization $x$ and its time derivative $\dot{X}_t$.

**Definition A.1. [23]** The random variable $X_n$ converges to $x$ in the first-mean sense (limit in mean) if for some $X$,

$$\lim_{n \to \infty} \mathbb{E}(|X_n - X|) = 0. \tag{A 1}$$

**Definition A.2.** The stochastic process $X_t$ is first-mean differentiable if for some $\dot{X}_t$

$$\lim_{\delta t \to 0} \mathbb{E} \left| \frac{X_{t+\delta t} - X_t}{\delta t} - \dot{X}_t \right| = 0. \tag{A 2}$$

**Definition A.3.** For given random variable $X$, the moment generating function (MGF) is defined by

$$\Phi_X(t) = E[\exp(Xt)] = \int_{-\infty}^{\infty} \exp(xt)\rho(x)dx. \tag{A 3}$$

**Proposition A.4.** *If $\Phi_X(t)$ is the MGF, then*

1. $d\Phi_X/dt|_{t=0} = m$, *where $m$ is the moment of $X$.*
2. *Let $X$ and $Y$ be two random variables. $X$ and $Y$ have the same distribution if and only if they have the same MGFs.*
3. *We say $X \sim N(\mu, \eta^2)$ if and only if $\Phi_X(t) = \exp^{(\eta^2 t^2/2) + \mu t}$.*
4. *If $X$ and $Y$ are two random variables, then the MGF $\Phi_{X+Y}(t) = \Phi_X(t)\Phi_Y(t)$.*

By the above propositions, one has

**Lemma A.5.** *If $X$, $Y$ are two independent Gaussian random variables with means $\mu_X$, $\mu_Y$ and covariances $\eta_X^2$, $\eta_Y^2$, then $X + Y$ is a Gaussian random variable with mean $\mu_X + \mu_y$ and covariance $\eta_X^2 + \eta_y^2$.*

**Definition A.6. [19]** A real-valued stochastic process $\{X_t\}_{t \in T}$, where $T$ is an index set, is a Gaussian process if all the finite-dimensional distributions are a multivariate normal distribution. That is, for any choice of distinct values $t_1, t_2, \ldots t_N \in T$, the random vector $X = (X_{t_1}, \ldots, X_{t_N})^T$ has a multivariate normal distribution with joint Gaussian probability density function given by

$$\rho_{X_{t_1} X_{t_2} \ldots X_{t_N}}(x_{t_1}, \ldots, x_{t_N}) = \frac{1}{(2\pi)^{N/2} \det(\Sigma)^{1/2}} \exp\left(-\frac{1}{2}(x - \mu_X)^T \Sigma^{-1}(x - \mu_X)\right). \tag{A 4}$$

where the mean vector is defined as

$$(\mu_X)_i = \mathbb{E}[X_{t_i}], \tag{A 5}$$

and covariance matrix $(\Sigma)_{ij} = \text{cov}(X_{t_i}, X_{t_j})$.

The Gaussian processes only depend on the mean and covariance functions. Usual covariance functions could be squared exponential $\text{cov}(X_{t_i}, X_{t_j}) = k_\phi(t_i, t_j) = \exp(-(1/2l^2)|t_i - t_j|^2)$, where $l$ is a hyperparameter and represents the non-local interaction length scale.

Let $t_0$, $\delta t \in R$ and $X_t$ be a Gaussian process with constant mean $\mu$ and kernel function $k_\phi(t_1, t_2)$, assumed to be first-mean differentiable. Then $X_{t_0+\delta t}$ and $X_{t_0}$ are jointly Gaussian distributed

$$\begin{bmatrix} X_{t_0} \\ X_{t_0+\delta t} \end{bmatrix} \sim \mathcal{N}\left(\begin{bmatrix} \mu \\ \mu \end{bmatrix}, \Sigma\right) \tag{A 6}$$

with density function

$$\rho(x_{t_0}, x_{t_0+\delta t}) = \frac{1}{2\pi \det(\Sigma)^{1/2}} \exp\left(-\frac{1}{2}\begin{bmatrix} x_{t_0} - \mu \\ x_{t_0+\delta t} - \mu \end{bmatrix}^T \Sigma^{-1} \begin{bmatrix} x_{t_0} - \mu \\ x_{t_0+\delta t} - \mu \end{bmatrix}\right), \tag{A 7}$$

where

$$\boldsymbol{\Sigma} = \begin{pmatrix} k_\phi(t_0, t_0) & k_\phi(t_0, t_0 + \delta t) \\ k_\phi(t_0 + \delta t, t_0) & k_\phi(t_0 + \delta t, t_0 + \delta t) \end{pmatrix}. \tag{A 8}$$

If we define linear transformation

$$\mathbf{T} = \begin{pmatrix} 1 & 0 \\ -\dfrac{1}{\delta t} & \dfrac{1}{\delta t} \end{pmatrix}, \tag{A 9}$$

then we have

$$\begin{bmatrix} X_{t_0} \\ \frac{X_{t_0+\delta t} - X_{t_0}}{\delta t} \end{bmatrix} = \mathbf{T} \begin{bmatrix} X_{t_0} \\ X_{t_0+\delta t} \end{bmatrix} \sim \mathcal{N}\left( \begin{bmatrix} \mu \\ 0 \end{bmatrix}, \mathbf{T}\boldsymbol{\Sigma}\mathbf{T}^{\mathrm{T}} \right), \tag{A 10}$$

i.e.

$$\rho(X_{t_0}, \frac{X_{t_0+\delta t} - X_{t_0}}{\delta t}) = \mathcal{N}\left( \begin{bmatrix} \mu \\ 0 \end{bmatrix}, \mathbf{T}\boldsymbol{\Sigma}\mathbf{T}^{\mathrm{T}} \right), \tag{A 11}$$

where

$$\mathbf{T}\boldsymbol{\Sigma}\mathbf{T}^{\mathrm{T}} = \begin{pmatrix} k_\phi(t_0, t_0) & \frac{k_\phi(t_0, t_0+\delta t) - k_\phi(t_0, t_0)}{\delta t_0} \\ \frac{k_\phi(t_0+\delta t_0, t_0) - k_\phi(t_0, t_0)}{\delta t} & \frac{k_\phi(t_0+\delta t_0, t_0+\delta t) - k_\phi(t_0, t_0+\delta t)}{\delta t_0} - \frac{k_\phi(t_0+\delta t, t_0) - k_\phi(t_0, t_0)}{\delta t_0} \end{pmatrix}. \tag{A 12}$$

The above derivation shows that $X_{t_0}$ and $(X_{t_0+\delta t} - X_{t_0})/\delta t$ are jointly Gaussian distributed. Using the definition of *first-mean* differential and the fact that $r$th-mean convergence implies convergence in distribution, it is clear that $X_{t_0}$ and $\dot{X}_{t_0}$ are jointly Gaussian

$$\begin{bmatrix} X_{t_0} \\ \dot{X}_{t_0} \end{bmatrix} \sim \mathcal{N}\left( \begin{bmatrix} \mu \\ 0 \end{bmatrix}, \begin{bmatrix} k_\phi(t_0, t_0) & \frac{\partial k_\phi(a, b)}{\partial b}\big|_{a=t_0, b=t_0} \\ \frac{\partial k_\phi(a, b)}{\partial a}\big|_{a=t_0, b=t_0} & \frac{\partial^2 k_\phi(a, b)}{\partial a \partial b}\big|_{a=t_0, b=t_0} \end{bmatrix} \right). \tag{A 13}$$

In general, $\boldsymbol{X} = (X_{t_1}, \ldots, X_{t_k})^{\mathrm{T}}$ and $\dot{\boldsymbol{X}} = (\dot{X}_{t_1}, \ldots, \dot{X}_{t_k})^{\mathrm{T}}$ are jointly Gaussian

$$\begin{bmatrix} \boldsymbol{X} \\ \dot{\boldsymbol{X}} \end{bmatrix} \sim \mathcal{N}\left( \begin{bmatrix} \boldsymbol{\mu} \\ \mathbf{0} \end{bmatrix}, \begin{bmatrix} \boldsymbol{C}_\phi(\boldsymbol{X}, \boldsymbol{X}) & \boldsymbol{C}_\phi(\boldsymbol{X}, \dot{\boldsymbol{X}}) \\ \boldsymbol{C}_\phi(\dot{\boldsymbol{X}}, \boldsymbol{X}) & \boldsymbol{C}_\phi(\dot{\boldsymbol{X}}, \dot{\boldsymbol{X}}) \end{bmatrix} \right). \tag{A 14}$$

Here $(C_\phi(\mathbf{a}, \mathbf{b}))_{ij} = cov(a_i, b_j)$ is the covariance between $a_i$ and $b_j$, and predefined kernel matrix of Gaussian process. By linearity of the covariance operator and the predefined kernel function $k_\phi(a, b)$, we have

$$C_\phi(X_{t_i}, \dot{X}_{t_j}) = \frac{\partial k_\phi(a, b)}{\partial b}\big|_{a=t_i, b=t_j}, \tag{A 15}$$

$$C_\phi(\dot{X}_{t_i}, X_{t_j}) = \frac{\partial k_\phi(a, b)}{\partial a}\big|_{a=t_i, b=t_j} \tag{A 16}$$

and

$$C_\phi(\dot{X}_{t_i}, \dot{X}_{t_j}) = \frac{\partial^2 k_\phi(a, b)}{\partial a \partial b}\big|_{a=t_i, b=t_j}. \tag{A 17}$$

**Lemma A.7. (Matrix Inversions Lemma)** *Let $\boldsymbol{\Sigma}$ be a $p \times p$ matrix ($p = n + m$):*

$$\boldsymbol{\Sigma} = \begin{bmatrix} \boldsymbol{\Sigma}_{11} & \boldsymbol{\Sigma}_{12} \\ \boldsymbol{\Sigma}_{21} & \boldsymbol{\Sigma}_{22} \end{bmatrix}, \tag{A 18}$$

*where the sum matrices have dimension $n \times n$, $n \times m$, etc. Suppose $\boldsymbol{\Sigma}$, $\boldsymbol{\Sigma}_{11}$, $\boldsymbol{\Sigma}_{22}$ are non-singular; and partition the inverse in the same way as $\boldsymbol{\Sigma}$,*

$$\boldsymbol{\Lambda} = \boldsymbol{\Sigma}^{-1} = \begin{bmatrix} \boldsymbol{\Lambda}_{11} & \boldsymbol{\Lambda}_{12} \\ \boldsymbol{\Lambda}_{21} & \boldsymbol{\Lambda}_{22} \end{bmatrix}. \tag{A 19}$$

*Then*

$$\begin{aligned}
\mathbf{\Lambda}_{11} &= (\mathbf{\Sigma}_{11} - \mathbf{\Sigma}_{12}\mathbf{\Sigma}_{22}^{-1}\mathbf{\Sigma}_{21})^{-1} \\
\mathbf{\Lambda}_{12} &= -(\mathbf{\Sigma}_{11} - \mathbf{\Sigma}_{12}\mathbf{\Sigma}_{22}^{-1}\mathbf{\Sigma}_{21})^{-1}\mathbf{\Sigma}_{12}\mathbf{\Sigma}_{22}^{-1}, \\
\mathbf{\Lambda}_{21} &= -(\mathbf{\Sigma}_{22} - \mathbf{\Sigma}_{21}\mathbf{\Sigma}_{11}^{-1}\mathbf{\Sigma}_{12})^{-1}\mathbf{\Sigma}_{21}\mathbf{\Sigma}_{11}^{-1} \\
\mathbf{\Lambda}_{22} &= (\mathbf{\Sigma}_{22} - \mathbf{\Sigma}_{21}\mathbf{\Sigma}_{11}^{-1}\mathbf{\Sigma}_{12})^{-1}.
\end{aligned} \right\} \tag{A 20}$$

*and*

**Lemma A.8. (conditional Gaussian distributions)** *Let $X \in \mathbb{R}^D$, $Y \in \mathbb{R}^M$, be jointly Gaussian random vectors with distribution*

$$\begin{bmatrix} X \\ Y \end{bmatrix} \sim \mathcal{N}(\boldsymbol{\mu}, \mathbf{\Sigma}), \tag{A 21}$$

*where*

$$\boldsymbol{\mu} = \begin{bmatrix} \boldsymbol{\mu}_X \\ \boldsymbol{\mu}_Y \end{bmatrix}, \mathbf{\Sigma} = \begin{bmatrix} \mathbf{\Sigma}_{XX} & \mathbf{\Sigma}_{XY} \\ \mathbf{\Sigma}_{YX} & \mathbf{\Sigma}_{YY} \end{bmatrix}. \tag{A 22}$$

*Then the conditional Gaussian distributions density functions are*

$$\rho_{Y|X}(y|x) = \frac{\rho_{XY}(x,y)}{\rho_X(x)} = \frac{1}{(2\pi)^{(M+D/2)}\det(\mathbf{\Sigma}_{Y|X})^{1/2}}\exp(y - \boldsymbol{\mu}_{Y|X})^{\mathrm{T}}\mathbf{\Sigma}_{Y|X}^{-1}(y - \boldsymbol{\mu}_{Y|X}) \tag{A 23}$$

*where*

$$\boldsymbol{\mu}_{Y|X} = \boldsymbol{\mu}_Y + \mathbf{\Sigma}_{YX}\mathbf{\Sigma}_{XX}^{-1}(x - \boldsymbol{\mu}_X) \tag{A 24}$$

*and*

$$\mathbf{\Sigma}_{Y|X} = \mathbf{\Sigma}_{YY} - \mathbf{\Sigma}_{YX}\mathbf{\Sigma}_{XX}^{-1}\mathbf{\Sigma}_{XY}. \tag{A 25}$$

According to above lemma, we have the condition distribution

**Lemma A.9.**

$$\rho(\dot{x}|x) \sim \mathcal{N}(D(x - \boldsymbol{\mu}_X), A), \tag{A 26}$$

*where*

$$D = C_\phi(\dot{X}, X)C_\phi(X, X)^{-1} \tag{A 27}$$

*and*

$$A = C_\phi(\dot{X}, \dot{X}) - C_\phi(\dot{X}, X)C_\phi(X, X)^{-1}C_\phi(X, \dot{X}). \tag{A 28}$$

# Appendix B. Proof of theorem 2.1

*Proof.* The joint density over all variables in figure 1 can be represented as

$$\rho(x_M, \dot{x}_M, y, x_L, f_M, \bar{f}_M, \boldsymbol{\theta}|\phi, \sigma, \gamma) \tag{B 1}$$
$$= \rho_{GP}(x_M, \dot{x}_M, y|\phi, \sigma)\rho_{ODE}(f_M, \bar{f}_M, \theta, x_L|x_M, \dot{x}_M, \gamma)$$
$$\rho_{GP}(x_M, \dot{x}_M, y|\phi, \sigma) = \rho(x_M|\phi)\rho(y|x_M, \sigma)\rho(\dot{x}_M|x_M, \phi) \tag{B 2}$$
$$\rho_{ODE}(f_M, \bar{f}_M, \theta, x_L|x_M, \dot{x}_M, \gamma)$$
$$= \rho(\boldsymbol{\theta})\rho(f_M, \bar{f}_M, x_L|\boldsymbol{\theta}, x_M, \dot{x}_M, \gamma)$$
$$= \rho(\boldsymbol{\theta})\rho(x_L|\boldsymbol{\theta}, x_M)\rho(f_M, \bar{f}_M|x_L, \boldsymbol{\theta}, x_M, \dot{x}_M, \gamma)$$
$$= \rho(\boldsymbol{\theta})\delta(\tilde{x}_L(\boldsymbol{\theta}, x_M) - x_L)\rho(f_M, \bar{f}_M|x_L, \boldsymbol{\theta}, x_M, \dot{x}_M, \gamma) \tag{B 3}$$
$$= \rho(\boldsymbol{\theta})\rho(f_M|\boldsymbol{\theta}, x_M, \tilde{x}_L(x_M, \boldsymbol{\theta}))\rho(\bar{f}_M|\dot{x}_M, \gamma)\delta(f_M - \bar{f}_M)$$
$$= \rho(\boldsymbol{\theta})\delta(f_M(\boldsymbol{\theta}, x_M, \tilde{x}_L(x_M, \boldsymbol{\theta})) - f_M)\mathcal{N}(f_M|\dot{x}_M, \gamma I)$$
$$= \rho(\boldsymbol{\theta})\mathcal{N}(f_M(\boldsymbol{\theta}, x_M, \tilde{x}_L(x_M, \boldsymbol{\theta}))|\dot{x}_M, \gamma I),$$

by which $\rho_{\mathrm{ODE}}$ is independent of $f_M$, $\bar{f}_M$, $x_L$. $\tilde{x}_L$ is deterministically decided by $x_M$, $\boldsymbol{\theta}$ through integration. Using lemma A.9, we have

$$\rho(x_M, \boldsymbol{\theta}, y | \phi, \sigma, \gamma) = \rho(\boldsymbol{\theta})\mathcal{N}(x_M | 0, C_\phi)\mathcal{N}(y | x_M, \sigma^2 I)\mathcal{N}(\dot{x}_M | Dx_M, A)\mathcal{N}(f_M(\boldsymbol{\theta}, x_M, \tilde{x}_L(x_M, \boldsymbol{\theta})) | \dot{x}_M, \gamma I). \quad (B\,4)$$

Integrating $\dot{x}_M$ out yields

$$\rho(x_M, \boldsymbol{\theta}, y | \phi, \sigma, \gamma) = \rho(\boldsymbol{\theta})\mathcal{N}(x_M | 0, C_\phi)\mathcal{N}(y | x_M, \sigma^2 I)\mathcal{N}(f_M(\boldsymbol{\theta}, x_M, \tilde{x}_L(x_M, \boldsymbol{\theta})) | Dx_M, A + \gamma I). \quad (B\,5)$$

Finally, we get

$$\rho(x_M, \boldsymbol{\theta} | y, \phi, \sigma, \gamma) \propto \rho(\boldsymbol{\theta})\mathcal{N}(x_M | 0, C_\phi)\mathcal{N}(y | x_M, \sigma^2 I)\mathcal{N}(f_M(\boldsymbol{\theta}, x_M, \tilde{x}_L(x_M, \boldsymbol{\theta})) | Dx_M, A + \gamma I). \quad (B\,6)$$

∎

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
