## [Peer Review File · Royal Society Open Science]

Review History

RSOS-200932.R0 (Original submission)

Review form: Reviewer 1

Is the manuscript scientifically sound in its present form?

Yes

Are the interpretations and conclusions justified by the results?

Yes

Is the language acceptable?

Yes

Do you have any ethical concerns with this paper?

No

Have you any concerns about statistical analyses in this paper?

No

Recommendation?

Accept as is

Comments to the Author(s)

This paper proposed an algorithm for parameter inference of coupled ODE systems with partially observable data. This algorithm combined a Gaussian process based gradient matching and a least square optimisation.

This is a good paper, well written and very clear in its findings. I believe that the Journal of the Royal Society Open Science is a good location for its publication.

Review form: Reviewer 2**Is the manuscript scientifically sound in its present form?**

No

Are the interpretations and conclusions justified by the results?

No

Is the language acceptable?

Yes

Do you have any ethical concerns with this paper?

No

Have you any concerns about statistical analyses in this paper?

No

Recommendation?

Reject

Comments to the Author(s)

The ScholarOne online submission system seems to remove all line breaks from standard text; hence my apologies for the following unformatted text.

SUMMARY:

The topic of the manuscript is "fast" parameter estimation in ODEs using gradient matching with Gaussian processes (GPs). The authors' new algorithm adds two modifications to related existing algorithms from the literature:

- 1) it combines an MCMC-based sampling scheme with deterministic optimization;
- 2) it can deal with partial observations by imputing missing values based on numerical integration of the ODEs.

The proposed algorithm is evaluated on three benchmark systems widely used in the related literature.

EVALUATION:

The authors demonstrate comprehensive knowledge of the relevant literature, and the mathematical derivations of their algorithm are sound.

However, there are four fundamental problems with the paper.

1) Imputation of missing values with GP-based gradient matching isn't new; it has, for instance, already been proposed in Reference [1] (see Sections 4 and 5.3). As opposed to the method in [1], the approach proposed in the submitted manuscript requires repeated numerical integrations of the ODEs. This is NOT an innovation, it is a clear disadvantage! The whole idea of gradient matching is to bypass the computationally expensive numerical integration of the ODEs. That won't become apparent in the applications chosen by the authors, because their toy problems are relatively simple and there is no need for gradient matching (and hence the algorithm proposed by the authors) in the first place. However, when dealing with complex systems and ODEs that are computationally expensive to solve numerically, the authors' imputation step will be a serious disadvantage over existing proper gradient-matching imputation methods like [1].

To explain this differently, it is difficult to see where the algorithm proposed by the authors would become relevant. If the system of ODEs is so simple that repeated numerical integrations are computationally feasible, then there is no need for any approximation based on gradient matching, and the authors' method becomes obsolete. If, on the other hand, the system of ODEs is so complex that repeated numerical integrations are practically not feasible - which is the very motivation for gradient matching - then the authors' imputation step is practically not feasible either.

2) The combination of MCMC with a follow-up optimization step can hardly be regarded as innovative. Approximately sampling parameters from an approximate posterior distribution properly quantifies uncertainty; parameter optimization loses this attractive feature. The advantage of optimization over MCMC-based sampling is the lower computational cost. However, to first invest computational resources to run MCMC simulations, and then ditch their main asset (uncertainty quantification) to reduce the result to a point estimate based on a follow-up optimization is counter-intuitive, counter-productive, and methodologically absurd.

3) The authors have tested their method on simple toy problems. One could argue that the computational costs of numerically integrating the ODEs are so low here that no gradient matching scheme, like the one proposed by the authors, is needed. However, in fairness to the authors, one has to acknowledge that most other publications on this topic use the same toy problems. This is okay, as long as computational complexity is properly quantified in terms of forward simulations from the model (which can be generalized to other more complex ODE systems). The problem, as pointed out above, is that the authors seem to assume that because their toy problems are computationally so cheap, repeated numerical integrations of the ODEs, as required for their imputation steps, are not an issue. This is totally misleading, in that their algorithm won't be applicable to more complex systems.

4) An advantage of standard benchmark toy problems, like the ones used by the authors, is that they have been widely used by other authors, enabling a comparison with related methods from the literature. It is peculiar that the authors of the submitted manuscript, despite demonstrating sound knowledge of the relevant literature, have not attempted a comparison of their method with any other existing method. In particular, the comparison with the method from Reference [1] is conspicuous by its absence.

CONCLUSION:

Related methods in the literature, like [1], can be applied to computationally complex ODE systems and provide a natural way of uncertainty quantification. The method proposed in the submitted manuscript loses both of these important features, and it has not even been compared with other state-of-the-art methods.

With no relevant methodological innovation and no comparative evaluation with alternative state-of-the-art methods, I don't think that the submitted manuscript has sufficient merit for a journal publication.

MINOR COMMENTS:

Page 5, Figure 1:

This is not a probabilistic graphical model (which by definition is a directed acyclic graph), but a chain graph (because the edge between FM and FarM is undirected). See reference [2] for details.

Page 5, equation (8):

This is not an equality, but a proportionality relationship.

Eq. (50):

$\rho(x_M, \phi) \rightarrow \rho(x_M | \phi)$

REFERENCES:

[1] Ben Calderhead, Mark Girolami and Neil D. Lawrence. Accelerating Bayesian inference over nonlinear differential equations with Gaussian processes. Neural Information Processing Systems (NIPS), 2008.

[2] D. Barber, Y. Wang. Gaussian processes for Bayesian estimation in ordinary differential equations. International Conference on Machine Learning, 2014

Review form: Reviewer 3

Is the manuscript scientifically sound in its present form?

Yes

Are the interpretations and conclusions justified by the results?

No

Is the language acceptable?

Yes

Do you have any ethical concerns with this paper?

No

Have you any concerns about statistical analyses in this paper?

Yes

Recommendation?

Major revision is needed (please make suggestions in comments)

Comments to the Author(s)

In their manuscript entitled "Numerical Method for Parameter Inference of Nonlinear ODEs with Partial Observations" the authors propose an extension to the known FGPGM framework by adding a regularization term.

In the provided version of the manuscript needs major improvement to figure out the novelty and the benefit of the introduced regularization term. Currently it looks like a summary of the paper from Wenk et al.

It is not clear to me if the method can be applied to all ODE systems or just to biological systems. Given the introduction and the title it can be applied to all types of ODE systems. In that case the authors should add one or two non-biological examples. Otherwise Title, Abstract and Introduction have to be changed. To clarify further, the title should be changed to ".. Parameter Inference of SYSTEMS OF Nonlinear ODEs... ". Moreover typical a major challenge with biological systems is their incompleteness, the fact that the states can not observed directly and that initial parameters are obtained from different experimental settings.

Page 3 Line 27: Please explain what is meant with "two different approaches"

Page 3 Line 54: As the authors may know there are different types of parameter identifiability. This should be figured out and explained in detail. Can the approach deal with all types of identifiability?

Page 3 Line 43 Word 2: Typo. The authors should carefully review their manuscript.

Page 4 Line 12: What do the authors mean with this sentence?

The following section describes the Algorithm similar to Weng et al. This is fine but the authors should mention this at the beginning of Section 2.

Section 2: For the benefit of the reader the authors should either include an index for the time-points or explain how the algorithm translates to a dynamic system. Especially Figure 1; which is reproduced from Weng et al; should be translated into a dynamic framework and/or the original authors have to be mentioned here.

Section 2: What about more complex systems where only combination of states can be estimated: $y=h(x)+e$. This is the usual case in biological systems. Please explain and clarify.

Section 2: Again how does the framework and the algorithm on Page 6 translate to a time dynamic system in detail? This is crucial for the understanding of the framework.

Section 2 Algorithm 1: How can the initial parameters are derived in detail. What is GP? Gaussian Process? Why do you set tau equal empty in the second FOR loop. Looks like this is not correct.

Section 2: As I understand from this section you add a L2 norm to the framework. Here time is introduced with T. Please explain further. Also, what are the assumptions and limitations by introducing this kind of regularization and how do boundary values affect the framework? This needs more explanation.

In Section 3 the authors reproduce the Experiments by Weng et al. As mentioned non-biological examples are missing.

More important, I did not found any supplementary code or ready-to use examples. Was this done in Matlab? I feel in the case of this work the code and examples needs to be published, as well. Especially for a final review the code is needed!

Page 7 Line 36... "refer FGPGM to the adapted FGPGM" this sounds odd and should be changed.

There are too much tables with confusing numbers. The authors should consider to generate head-maps instead. (For example Table 1/Sensitivity tables)

I miss some comments regarding the running time and robustness and impact of the link function $h(y=h(x)+e)$. Also Page 13 line 33.. "Thus a least square optimization after doing FGPGM may well reduce this effect (of smoother trajectories)". Isn't the opposite the case after applying the L2 norm?

In the current version the discussion is very weak and needs improvement, also Page 18 Line 53 is not supported by the results. How does the dimension of the ODE system translate into computational costs and accuracy? In practice systems have more than 10 Equations and species.

Decision letter (RSOS-200932.R0)

Dear Dr Xu

The Editors assigned to your paper RSOS-200932 "Numerical Method for Parameter Inference of Nonlinear ODEs with Partial Observations" have made a decision based on their reading of the paper and any comments received from reviewers.

Regrettably, in view of the reports received, the manuscript has been rejected in its current form. However, a new manuscript may be submitted which takes into consideration these comments.

We invite you to respond to the comments supplied below and prepare a resubmission of your manuscript. Below the referees' and Editors' comments (where applicable) we provide additional requirements. We provide guidance below to help you prepare your revision.

Please note that resubmitting your manuscript does not guarantee eventual acceptance, and we do not generally allow multiple rounds of revision and resubmission, so we urge you to make every effort to fully address all of the comments at this stage. If deemed necessary by the Editors, your manuscript will be sent back to one or more of the original reviewers for assessment. If the original reviewers are not available, we may invite new reviewers.

Please resubmit your revised manuscript and required files (see below) no later than 14-Feb-2021. Note: the ScholarOne system will 'lock' if resubmission is attempted on or after this deadline. If you do not think you will be able to meet this deadline, please contact the editorial office immediately.

Please note article processing charges apply to papers accepted for publication in Royal Society Open Science (<https://royalsocietypublishing.org/rsos/charges>). Charges will also apply to papers transferred to the journal from other Royal Society Publishing journals, as well as papers submitted as part of our collaboration with the Royal Society of Chemistry (<https://royalsocietypublishing.org/rsos/chemistry>). Fee waivers are available but must be requested when you submit your manuscript (<https://royalsocietypublishing.org/rsos/waivers>).

Thank you for submitting your manuscript to Royal Society Open Science and we look forward to receiving your resubmission. If you have any questions at all, please do not hesitate to get in touch.

on behalf of Professor Len Thomas (Associate Editor) and Mark Chaplain (Subject Editor)
openscience@royalsociety.org

Associate Editor Comments to Author (Professor Len Thomas):

Comments to the Author:

Thank-you for submitting to RSOS. I have now received three reviews, two of which make substantive criticisms of the manuscript. In the light of this, and my own reading of the paper, I am recommending rejection at this stage - but with the invitation to re-submit if the criticisms can be addressed. In particular, reviewer 2 points out that the proposed approach, requiring numerical integration, will be computationally expensive thereby negating the main advantage of gradient matching; they also suggest that the combination of MCMC plus optimization loses the attractive feature of MCMC that uncertainty can be readily quantified. Reviewer 2 points out that imputation has already been proposed by Calderhead et al (2008) and Reviewer 3 suggests there is little novel relative to Wenk et al. (2019).

If you do decide to revise and resubmit, I encourage you to deal with all of the points raised by the reviewers (not just those mentioned above). Reviewer 3 also points out that source code to reproduce the examples is required for acceptance at RSOS - and indeed it is required for further review, in line with our editorial policy. I believe the journal staff have contacted you about this already.

With best wishes, Len Thomas

Reviewer comments to Author:

Reviewer: 1

Comments to the Author(s)

This paper proposed an algorithm for parameter inference of coupled ODE systems with partially observable data. This algorithm combined a Gaussian process based gradient matching and a least square optimisation.

This is a good paper, well written and very clear in its findings. I believe that the Journal of the Royal Society Open Science is a good location for its publication.

Reviewer: 2

Comments to the Author(s)

The ScholarOne online submission system seems to remove all line breaks from standard text; hence my apologies for the following unformatted text.

SUMMARY:

The topic of the manuscript is "fast" parameter estimation in ODEs using gradient matching with Gaussian processes (GPs). The authors' new algorithm adds two modifications to related existing algorithms from the literature:

- 1) it combines an MCMC-based sampling scheme with deterministic optimization;
- 2) it can deal with partial observations by imputing missing values based on numerical integration of the ODEs.

The proposed algorithm is evaluated on three benchmark systems widely used in the related literature.

EVALUATION:

The authors demonstrate comprehensive knowledge of the relevant literature, and the mathematical derivations of their algorithm are sound.

However, there are four fundamental problems with the paper.

1) Imputation of missing values with GP-based gradient matching isn't new; it has, for instance, already been proposed in Reference [1] (see Sections 4 and 5.3). As opposed to the method in [1], the approach proposed in the submitted manuscript requires repeated numerical integrations of the ODEs. This is NOT an innovation, it is a clear disadvantage! The whole idea of gradient matching is to bypass the computationally expensive numerical integration of the ODEs. That won't become apparent in the applications chosen by the authors, because their toy problems are relatively simple and there is no need for gradient matching (and hence the algorithm proposed by the authors) in the first place. However, when dealing with complex systems and ODEs that are computationally expensive to solve numerically, the authors' imputation step will be a serious disadvantage over existing proper gradient-matching imputation methods like [1].

To explain this differently, it is difficult to see where the algorithm proposed by the authors would become relevant. If the system of ODEs is so simple that repeated numerical integrations are computationally feasible, then there is no need for any approximation based on gradient matching, and the authors' method becomes obsolete. If, on the other hand, the system of ODEs is so complex that repeated numerical integrations are practically not feasible - which is the very motivation for gradient matching - then the authors' imputation step is practically not feasible either.

2) The combination of MCMC with a follow-up optimization step can hardly be regarded as innovative. Approximately sampling parameters from an approximate posterior distribution properly quantifies uncertainty; parameter optimization loses this attractive feature. The advantage of optimization over MCMC-based sampling is the lower computational cost. However, to first invest computational resources to run MCMC simulations, and then ditch their main asset (uncertainty quantification) to reduce the result to a point estimate based on a follow-up optimization is counter-intuitive, counter-productive, and methodologically absurd.

3) The authors have tested their method on simple toy problems. One could argue that the computational costs of numerically integrating the ODEs are so low here that no gradient matching scheme, like the one proposed by the authors, is needed. However, in fairness to the authors, one has to acknowledge that most other publications on this topic use the same toy problems. This is okay, as long as computational complexity is properly quantified in terms of forward simulations from the model (which can be generalized to other more complex ODE systems). The problem, as pointed out above, is that the authors seem to assume that because their toy problems are computationally so cheap, repeated numerical integrations of the ODEs, as required for their imputation steps, are not an issue. This is totally misleading, in that their algorithm won't be applicable to more complex systems.

4) An advantage of standard benchmark toy problems, like the ones used by the authors, is that they have been widely used by other authors, enabling a comparison with related methods from the literature. It is peculiar that the authors of the submitted manuscript, despite demonstrating sound knowledge of the relevant literature, have not attempted a comparison of their method

with any other existing method. In particular, the comparison with the method from Reference [1] is conspicuous by its absence.

CONCLUSION:

Related methods in the literature, like [1], can be applied to computationally complex ODE systems and provide a natural way of uncertainty quantification. The method proposed in the submitted manuscript loses both of these important features, and it has not even been compared with other state-of-the-art methods.

With no relevant methodological innovation and no comparative evaluation with alternative state-of-the-art methods, I don't think that the submitted manuscript has sufficient merit for a journal publication.

MINOR COMMENTS:

Page 5, Figure 1:

This is not a probabilistic graphical model (which by definition is a directed acyclic graph), but a chain graph (because the edge between FM and FarM is undirected). See reference [2] for details.

Page 5, equation (8):

This is not an equality, but a proportionality relationship.

Eq. (50):

$\rho(x_M, \phi) \rightarrow \rho(x_M | \phi)$

REFERENCES:

[1] Ben Calderhead, Mark Girolami and Neil D. Lawrence.

Accelerating Bayesian inference over nonlinear differential equations with Gaussian processes. Neural Information Processing Systems (NIPS), 2008.

[2] D. Barber, Y. Wang. Gaussian processes for Bayesian estimation in ordinary differential equations. International Conference on Machine Learning, 2014

Reviewer: 3

Comments to the Author(s)

In their manuscript entitled "Numerical Method for Parameter Inference of Nonlinear ODEs with Partial Observations" the authors propose an extension to the known FGPGM framework by adding a regularization term.

In the provided version of the manuscript needs major improvement to figure out the novelty and the benefit of the introduced regularization term. Currently it looks like a summary of the paper from Wenk et al.

It is not clear to me if the method can be applied to all ODE systems or just to biological systems. Given the introduction and the title it can be applied to all types of ODE systems. In that case the authors should add one or two non-biological examples. Otherwise Title, Abstract and Introduction have to be changed. To clarify further, the title should be changed to ".. Parameter Inference of SYSTEMS OF Nonlinear ODEs... ". Moreover typical a major challenge with biological systems is their incompleteness, the fact that the states can not be observed directly and that initial parameters are obtained from different experimental settings.

Page 3 Line 27: Please explain what is meant with "two different approaches"

Page 3 Line 54: As the authors may know there are different types of parameter identifiability. This should be figured out and explained in detail. Can the approach deal with all types of identifiability?

Page 3 Line 43 Word 2: Typo. The authors should carefully review their manuscript.

Page 4 Line 12: What do the authors mean with this sentence?

The following section describes the Algorithm similar to Weng et al. This is fine but the authors should mention this at the beginning of Section 2.

Section 2: For the benefit of the reader the authors should either include an index for the time-points or explain how the algorithm translates to a dynamic system. Especially Figure 1; which is reproduced from Weng et al; should be translated into a dynamic framework and/or the original authors have to be mentioned here.

Section 2: What about more complex systems where only combination of states can be estimated: $y=h(x)+e$. This is the usual case in biological systems. Please explain and clarify.

Section 2: Again how does the framework and the algorithm on Page 6 translate to a time dynamic system in detail? This is crucial for the understanding of the framework.

Section 2 Algorithm 1: How can the initial parameters are derived in detail. What is GP? Gaussian Process? Why do you set tau equal empty in the second FOR loop. Looks like this is not correct.

Section 2: As I understand from this section you add a L2 norm to the framework. Here time is introduced with T. Please explain further. Also, what are the assumptions and limitations by introducing this kind of regularization and how do boundary values affect the framework? This needs more explanation.

In Section 3 the authors reproduce the Experiments by Weng et al. As mentioned non-biological examples are missing.

More important, I did not found any supplementary code or ready-to use examples. Was this done in Matlab? I feel in the case of this work the code and examples needs to be published, as well. Especially for a final review the code is needed!

Page 7 Line 36... "refer FGPGM to the adapted FGPGM" this sounds odd and should be changed.

There are too much tables with confusing numbers. The authors should consider to generate head-maps instead. (For example Table 1/Sensitivity tables)

I miss some comments regarding the running time and robustness and impact of the link function $h(y=h(x)+e)$. Also Page 13 line 33.. "Thus a least square optimization after doing FGPGM may well reduce this effect (of smoother trajectories)". Isn't the opposite the case after applying the L2 norm?

In the current version the discussion is very weak and needs improvement, also Page 18 Line 53 is not supported by the results. How does the dimension of the ODE system translate into computational costs and accuracy? In practice systems have more than 10 Equations and species.

===PREPARING YOUR MANUSCRIPT===

===PREPARING YOUR REVISION IN SCHOLARONE===

- An editable file of each table (.doc, .docx, .xls, .xlsx, or .csv).
- An editable file of all figure and table captions.

- Any electronic supplementary material (ESM).
- If you are requesting a discretionary waiver for the article processing charge, the waiver form must be included at this step.
- If you are providing image files for potential cover images, please upload these at this step, and inform the editorial office you have done so. You must hold the copyright to any image provided.
- A copy of your point-by-point response to referees and Editors. This will expedite the preparation of your proof.

- Ensure that your data access statement meets the requirements at <https://royalsociety.org/journals/authors/author-guidelines/#data>. You should ensure that you cite the dataset in your reference list. If you have deposited data etc in the Dryad repository, please include both the 'For publication' link and 'For review' link at this stage.
- If you are requesting an article processing charge waiver, you must select the relevant waiver option (if requesting a discretionary waiver, the form should have been uploaded at Step 3 'File upload' above).
- If you have uploaded ESM files, please ensure you follow the guidance at <https://royalsociety.org/journals/authors/author-guidelines/#supplementary-material> to include a suitable title and informative caption. An example of appropriate titling and captioning may be found at [https://figshare.com/articles/Table_S2_from_Is_there_a_trade-off_between_peak_performance_and_performance_breadth_across_temperatures_for_aerobic_sc](https://figshare.com/articles/Table_S2_from_Is_there_a_trade-off_between_peak_performance_and_performance_breadth_across_temperatures_for_aerobic_scope_in_teleost_fishes_/3843624) ope_in_teleost_fishes_/3843624.

Author's Response to Decision Letter for (RSOS-200932.R0)

See Appendix A.

RSOS-210171.R0

Review form: Reviewer 3

Is the manuscript scientifically sound in its present form?

Yes

Are the interpretations and conclusions justified by the results?

No

Is the language acceptable?

Yes

Do you have any ethical concerns with this paper?

No

Have you any concerns about statistical analyses in this paper?

Yes

Recommendation?

Major revision is needed (please make suggestions in comments)

Comments to the Author(s)

The current version of the manuscript reads well, and the concept is much clearer now. Nevertheless, the authors should proofread their manuscript twice (e.g., page 6 line 23).

I really appreciate the accessible code to support this work.

In lines 46 and 53 on page 1 the authors make the point that they can deal with “partial observations” and “large noise”. I highly appreciate section 3.4 and other edits to detail this further. But still, I suggest including a better benchmark analysis. I suggest performing a proper analysis for example 3.3. I would like to see a table or AUC plot which demonstrates how the parameter error evolves with increasing noise and with exclusion of observations (sample size and complete dynamics/variables). This should be a straightforward exercise. Also, I recommend including the run-time. I am wondering how the run-time is related to the noise level.

Related to the comment above I would like to see how the algorithm performs on real data. Or alternatively in cases where the noise is not Gaussian. I feel the algorithm performs well on toy examples but not on real data. In addition, the authors should investigate the case of uncertain/unknown initial state conditions, as well. If this is not possible it has to be pointed out in the discussion that the performance with real data has to be further investigated.

The sentence on page 6 line 16 “This approach does not depend on the observed variables.” is hard to understand. The authors should consider rephrasing the sentence. It is a little bit confusing.

I recommend another revision to address remaining concerns, but the current manuscript shows high potential and fits very well to the journal.

Decision letter (RSOS-210171.R0)

Dear Dr Xu

On behalf of the Editors, we are pleased to inform you that your Manuscript RSOS-210171 "Numerical Method for Parameter Inference of Nonlinear ODEs with Partial Observations" has been accepted for publication in Royal Society Open Science subject to minor revision in accordance with the referees' reports. Please find the referees' comments along with any feedback from the Editors below my signature.

Please submit your revised manuscript and required files (see below) no later than 7 days from today's (ie 07-Jun-2021) date. Note: the ScholarOne system will 'lock' if submission of the revision is attempted 7 or more days after the deadline. If you do not think you will be able to meet this deadline please contact the editorial office immediately.

on behalf of Professor Len Thomas (Associate Editor) and Mark Chaplain (Subject Editor)
openscience@royalsociety.org

Associate Editor Comments to Author (Professor Len Thomas):

Comments to the Author:

Thank-you for your work on the manuscript to date. I feel it is almost ready for acceptance, however a reviewer is not happy with how well it may perform on real-world data. They write "I would like to see how the algorithm performs on real data. Or alternatively in cases where the noise is not Gaussian. I feel the algorithm performs well on toy examples but not on real data. In addition, the authors should investigate the case of uncertain/unknown initial state conditions, as well. If this is not possible it has to be pointed out in the discussion that the performance with real data has to be further investigated." Please could you either include a real-world example, or state explicitly that this needs further investigation.

The reviewer has made other minor comments that you should please address.

I do not anticipate your re-submission will need to go out for review again, and so we will be able to move forward quickly.

Reviewer comments to Author:

Reviewer: 3

Comments to the Author(s)

The current version of the manuscript reads well, and the concept is much clearer now. Nevertheless, the authors should proofread their manuscript twice (e.g., page 6 line 23).

I really appreciate the accessible code to support this work.

In lines 46 and 53 on page 1 the authors make the point that they can deal with “partial observations” and “large noise”. I highly appreciate section 3.4 and other edits to detail this further. But still, I suggest including a better benchmark analysis. I suggest performing a proper analysis for example 3.3. I would like to see a table or AUC plot which demonstrates how the parameter error evolves with increasing noise and with exclusion of observations (sample size and complete dynamics/variables). This should be a straightforward exercise. Also, I recommend including the run-time. I am wondering how the run-time is related to the noise level.

Related to the comment above I would like to see how the algorithm performs on real data. Or alternatively in cases where the noise is not Gaussian. I feel the algorithm performs well on toy examples but not on real data. In addition, the authors should investigate the case of uncertain/unknown initial state conditions, as well. If this is not possible it has to be pointed out in the discussion that the performance with real data has to be further investigated.

The sentence on page 6 line 16 “This approach does not depend on the observed variables.” is hard to understand. The authors should consider rephrasing the sentence. It is a little bit confusing.

I recommend another revision to address remaining concerns, but the current manuscript shows high potential and fits very well to the journal.

===PREPARING YOUR MANUSCRIPT===

Your revised paper should include the changes requested by the referees and Editors of your manuscript. You should provide two versions of this manuscript and both versions must be provided in an editable format:
 one version identifying all the changes that have been made (for instance, in coloured highlight, in bold text, or tracked changes);
 a 'clean' version of the new manuscript that incorporates the changes made, but does not highlight them. This version will be used for typesetting.
 Please ensure that any equations included in the paper are editable text and not embedded images.

===PREPARING YOUR REVISION IN SCHOLARONE===

Author's Response to Decision Letter for (RSOS-210171.R0)

See Appendix B.

Decision letter (RSOS-210171.R1)

Dear Dr Xu,

I am pleased to inform you that your manuscript entitled "Numerical Method for Parameter Inference of Nonlinear ODEs with Partial Observations" is now accepted for publication in Royal Society Open Science.

on behalf of Professor Len Thomas (Associate Editor) and Mark Chaplain (Subject Editor)

Appendix A

We would like to thank all referees for their very helpful comments and suggestions. We extensively revised our paper to address all of them in detail. In what follows, we provide our responses to all individual comments and describe in detail specific changes in the revised manuscript made in response to the specific referees' comments and suggestions.

Responses

Reviewer: # 1

Comment #1 This paper proposed an algorithm for parameter inference of coupled ODE systems with partially observable data. This algorithm combined a Gaussian process based gradient matching and a least square optimization. This is a good paper, well written and very clear in its findings. I believe that the Journal of the Royal Society Open Science is a good location for its publication.

Response: We thank the reviewer for the positive evaluation of our paper.

Reviewer: 2

Comments to the Author(s)

Comment #1 The Scholar One online submission system seems to remove all line breaks from standard text; hence my apologies for the following unformatted text.

SUMMARY:

The topic of the manuscript is "fast" parameter estimation in ODEs using gradient matching with Gaussian processes (GPs). The authors' new algorithm adds two modifications to related existing algorithms from the literature:

- 1) it combines an MCMC-based sampling scheme with deterministic optimization;
- 2) it can deal with partial observations by imputing missing values based on numerical integration of the ODEs.

The proposed algorithm is evaluated on three benchmark systems widely used in the related literature.

2) EVALUATION:

The authors demonstrate comprehensive knowledge of the relevant literature, and the mathematical derivations of their algorithm are sound.

However, there are four fundamental problems with the paper.

- 1) Imputation of missing values with GP-based gradient matching isn't new; it has, for instance, already been proposed in Reference [1] (see Sections 4 and 5.3). As opposed to the method in [1], the approach proposed in the submitted manuscript requires repeated numerical

integrations of the ODEs. This is NOT an innovation, it is a clear disadvantage! The whole idea of gradient matching is to bypass the computationally expensive numerical integration of the ODEs. That won't become apparent in the applications chosen by the authors, because their toy problems are relatively simple and there is no need for gradient matching (and hence the algorithm proposed by the authors) in the first place. However, when dealing with complex systems and ODEs that are computationally expensive to solve numerically, the authors' imputation step will be a serious disadvantage over existing proper gradient-matching imputation methods like [1].

To explain this differently, it is difficult to see where the algorithm proposed by the authors would become relevant. If the system of ODEs is so simple that repeated numerical integrations are computationally feasible, then there is no need for any approximation based on gradient matching, and the authors' method becomes obsolete. If, on the other hand, the system of ODEs is so complex that repeated numerical integrations are practically not feasible - which is the very motivation for gradient matching - then the authors' imputation step is practically not feasible either.

Response: Thank you for your comments. In the revised manuscript, we compared three kinds of treatments of the unobserved variables, including the pros and cons.

1. Integrate the whole system, which is independent of the data of known variables but as referee pointed out it would be time consuming for large systems.
2. Partially integrate the unobservable variables. This approach saves time when only a few variables are unobservable but may need extra cost when interpolation and smoothing of the observed variables are needed.
3. Sample the unobserved variables with Gaussian process and do the gradient matching. The computational cost in each sampling cycle will be relatively low for large systems, but the result and convergence speed depends on initial guess.

According to our numerical results and discussion, we suggest that if the system is very small such that solving ODEs is cheap in time, one can adopt the integration approaches since they are independent of the prior information of the unobserved variables. If the integration requires fine time step and most of the variables in the system are unobservable, then the full integration is preferred since denoise and interpolation of the observable variables which appear in the equations for unobservable ones can be avoided. For large scale problems with reliable prior information, the full sampling method may have advantages in both accuracy and efficiency.

The related details and conclusions seems not found in Section 4 or Section 5.3 in reference [1]. From the following statement in [1] the smoothing of observed species and discrete equations are needed, which are only involved in the partial integration approach mentioned above rather than the sampling.

where $\delta_n^{o,u} \equiv \mathbf{f}_n(\mathbf{X}_o, \mathbf{X}_u, \boldsymbol{\theta}, \mathbf{t}) - \mathbf{m}_n$ and $\pi(\mathbf{X}_u)$ is an appropriately chosen prior. The values of unobserved species are obtained by propagating their sampled initial values using the corresponding discrete versions of the differential equations and the smoothed estimates of observed species. The

We hope our supplemented results can provide a more comprehensive understanding on the methods and the choice in practice.

Besides, we would like to clarify that the gradient matching may help avoid local minima, (minimize both data difference and time derivative difference). This shows advantage even for simple systems. As provided in Figure 9, the conventional least square method is trapped into a local minimum which doesn't happen for the gradient matching.

Comment #2 The combination of MCMC with a follow-up optimization step can hardly be regarded as innovative. Approximately sampling parameters from an approximate posterior distribution properly quantifies uncertainty; parameter optimization loses this attractive feature. The advantage of optimization over MCMC-based sampling is the lower computational cost. However, to first invest computational resources to run MCMC simulations, and then ditch their main asset (uncertainty quantification) to reduce the result to a point estimate based on a follow-up optimization is counter-intuitive, counter-productive, and methodologically absurd.

Response: Thank you very much for the comments. As we discussed above, the deterministic least square optimization may easily fall into local minimum points. One solution is to incorporate time derivative matching in the objective function. However, numerical differentiation is usually unstable and difficult task especially for noisy data. Gaussian process then provides a good way to deal with time derivative and realize gradient matching because it is closed under time differentiation. Then, the combination of them make advantages of the robustness of Gaussian process in dealing with time derivative and the low cost of optimization. Actually we do not need much MCMC samplings to obtain a good initial status for the optimization step.

Comment #3 The authors have tested their method on simple toy problems. One could argue that the computational costs of numerically integrating the ODEs are so low here that no gradient matching scheme, like the one proposed by the authors, is needed. However, in fairness to the authors, one has to acknowledge that most other publications on this topic use the same toy problems. This is okay, as long as computational complexity is properly quantified in terms of forward simulations from the model (which can be generalized to other more complex ODE systems). The problem, as pointed out above, is that the authors seem to assume that because their toy problems are computationally so cheap, repeated numerical integrations of the ODEs, as required for their imputation steps, are not an issue. This is totally misleading, in that their algorithm won't be applicable to more complex systems.

Response: Thank you for your comments. We agree that it is important to take into

consideration the computational cost for integration of the unobserved variables for complex systems. As illustrated above, in the revised manuscript we compared three approaches to infer the unobservable variables including sampling and integrations, and discussed the pros and cons and the choice in various situations based on the numerical results.

Comment #4 An advantage of standard benchmark toy problems, like the ones used by the authors, is that they have been widely used by other authors, enabling a comparison with related methods from the literature. It is peculiar that the authors of the submitted manuscript, despite demonstrating sound knowledge of the relevant literature, have not attempted a comparison of their method with any other existing method. In particular, the comparison with the method from Reference [1] is conspicuous by its absence.

Response: Thank you for the comments. We looked up the example involving unobserved variables used in Section 5.3 of [1]. However, the unobserved variable occurs as a time dependent function and there is no differential equation for that variable. This example is then not suitable for the approach that involving gradient matching for the unobserved variable (approach 3 mentioned above). For the known variables, both our methods and that in [1] apply gradient matching.

Comment #5 Page 5, Figure 1: This is not a probabilistic graphical model (which by definition is a directed acyclic graph), but a chain graph (because the edge between FM and FarM is undirected). See reference [2] for details.

Response: Thank you. It has been revised.

Figure 1: Chain graph with partial observable variables.

Comment #6 Page 5, equation (8):

Response: Thank you. It has been corrected.

Comment #7 This is not an equality, but a proportionality relationship. Eq. (50): $\rho(x_M, \phi) \rightarrow \rho(x_M | \phi)$

Response: It has been corrected (Eq. 51 in the revised manuscript).

Thank you again for your helpful comments and suggestions.

Reviewer: 3

Comment #1 In their manuscript entitled "Numerical Method for Parameter Inference of Nonlinear ODEs with Partial Observations" the authors propose an extension to the known FGPGM framework by adding a regularization term.

In the provided version of the manuscript needs major improvement to figure out the novelty and the benefit of the introduced regularization term. Currently it looks like a summary of the paper from Wenk et al.

Response: Thank you for the comment. The present method and numerical results focus on the case with unobserved variables, which is not considered in Wenk et al. (2019). The theoretical deduction, algorithms and examples are all extended for the situation with unobserved variables, which cannot be covered by the work of Wenk et al. (2019). Especially, we provide three approaches to deal with the unobserved variables and discuss based on numerical comparisons the pros and cons for each methods. Besides, we also proposed the combination of the gradient matching and least square optimization to take advantages of the former to prevent local minima and the later to reduce significantly the MCMC sampling numbers.

Comment #2 It is not clear to me if the method can be applied to all ODE systems or just to biological systems. Given the introduction and the title it can be applied to all types of ODE systems. In that case the authors should add one or two non-biological examples. Otherwise Title, Abstract and Introduction have to be changed. To clarify further, the title should be changed to "... Parameter Inference of SYSTEMS OF Nonlinear ODEs... ". Moreover typical a major challenge with biological systems is their incompleteness, the fact that the states can not observed directly and that initial parameters are obtained from different experimental settings.

Response: Thank you for the comment. In the revised version, we supplemented an example irrelevant to biological processes. We have added 'systems of' in the title. Actually, as we explained above, our main focus is the situation in the presence of unobserved variables.

Comment #3 Page 3 Line 27: Please explain what is meant with "two different approaches"

Response: The sentence has been changed as follows in the second paragraph of introduction

numerical integration [22, 7, 18, 11]. These techniques are based on minimizing the difference between the **time derivatives (gradients)** and the **right hand side of the equations**. This usually involves a process consisting of two steps: data interpolation and parameter adaptation. Among them, nonparametric Bayesian modelling with Gaussian processes is one of the promising approaches, which includes data interpolation by Gaussian process and parameter adaptation by matching the solution or model. An adaptive gradient

Comment #4 Page 3 Line 54: As the authors may know there are different types of parameter identifiability. This should be figured out and explained in detail. Can the approach deal with all types of identifiability?

Response: The identifiability is indeed an important topic. The current work does not involve the discussion of identifiability. We assume that theoretically the parameters in the examples are globally identifiable with the corresponding observations. This does not rule out the existence of local minima in the gradient matching or optimization. We have such example at the end of Section 3.3 to illustrate that the gradient matching may help avoid local minima which occurs in the least square optimization.

Comment #5 Page 3 Line 43 Word 2: Typo. The authors should carefully review their manuscript.

Response: Thank you for pointing that, we have corrected it and other typos.

Comment #6 Page 4 Line 12: What do the authors mean with this sentence?

Response: Thank you. We have changed this sentence into "We will provide three approaches to deal with the unobserved variables, including numerical integrations and Gaussian process sampling, and compare their performances."

Comment #7 The following section describes the Algorithm similar to Weng et al. This is fine but the authors should mention this at the beginning of Section 2.

Response: We are sorry about confusion. We have mentioned the work of Wenk et al at the beginning of Section 2.

The main strategy of Gaussian process based gradient matching is to minimize the mismatch between the data and the ODE solutions in a maximum likelihood sense, making use of the property that Gaussian process is closed under differentiation. In this section, we will extend the FGPGM method proposed in [25] to the situation that contains unobserved variables.

Actually, the formula and algorithm presented here are different from those in Wenk et al. and contains unobserved variables which cannot be covered by results in Wenk et al.

Comment #8 Section 2: For the benefit of the reader the authors should either include an index for the time-points or explain how the algorithm translates to a dynamic system. Especially Figure 1; which is reproduced from Weng et al; should be translated into a dynamic framework and/or the original authors have to be mentioned here.

Response: Thank you for the suggestion. The index for time-points are included in the corresponding locations (labeled in blue in Section2-paragraph 2).

In this work, we would like to estimate the time-independent parameters θ of the following dynamical system described by

$$\dot{X}(t) = f(X(t), \theta). \tag{1}$$

\dot{X} is the vector of time derivative of the variable $X = (X_1(t), X_2(t), \dots, X_{N_1}(t))$ and f can be an nonlinear vector valued functions. We assume only part of the variables are measurable and denote them as X_M . Throughout the paper we use subscript M to specify measurable components. They are observed on discrete time points as $Y(t_i)(i = 1, \dots, N_2)$ with noise ϵ such that $Y = X_M + \epsilon$. We assume that the noise is Gaussian $\epsilon(t_i) \sim \mathcal{N}(0, \sigma^2 I)$, then

$$\rho(y|x_M, \sigma) = \mathcal{N}(y|x_M, \sigma^2 I), \tag{2}$$

The difference between Figure 1 and the chain graph in Wenk et al., is the including of the unobserved variables.

Figure 1: Chain graph with partial observable variables.

Comment #9 Section 2: What about more complex systems where only combination of states can be estimated: $y=h(x)+e$. This is the usual case in biological systems. Please explain and clarify.

Response: Thank you for your suggestion. We have added an example with composite observer. Please see section 3.5 in the revised manuscript.

Comment #10 Section 2: Again how does the framework and the algorithm on Page 6 translate to a time dynamic system in detail? This is crucial for the understanding of the framework.

Response: Thank you. We have added indices for variables and discrete time points.

Comment #11 Section 2 Algorithm 1: How can the initial parameters are derived in detail. What is GP? Gaussian Process? Why do you set tau equal empty in the second FOR loop. Looks like this is not correct.

Response: The initial guess for parameters are chosen randomly within their value ranges. The ranges are empirically decided depending on specific problems and their orders of magnitude. For example, if physically a parameter should be positive and the true value is 1, then we can set the sampling range to be [0,5]. We have corrected the abbreviate and the position of initialization for tau, thank you.

```
Step 2. Infer  $x_M$ ,  $x_L$  and  $\theta$  using MCMC
 $S \leftarrow \emptyset$ 
for  $i = 1 \rightarrow N_{MCMC} + N_{burnin}$  do
   $\mathcal{T} \leftarrow \emptyset$ 
  for each state  $x_M(t_k)$  do
    Propose a new state value using a Gaussian distribution with standard
    deviation  $\sigma_x$ 
    Accept proposed value based on the density (Eq. 8)
    Add current value to  $\mathcal{T}$ 
  end for
  for each parameter do
    Propose a new parameter value using a Gaussian distribution with standard
    deviation  $\sigma_p$ 
    Infer  $x_L$  by integration or sampling.
    Accept proposed value based on the density (Eq. 8)
    Add current value to  $\mathcal{T}$ 
  end for
  Add the mean of  $\mathcal{T}$  to  $S$ 
end for
Discard the first  $N_{burnin}$  samples of  $S$ 
Return  $x_M$ ,  $x_L$ ,  $\theta$ 
Step 3. Optimization of Eq. 11 using  $\theta$  from Step 2 as initial guess.
```

Comment #12 Section 2: As I understand from this section you add a L2 norm to the framework. Here time is introduced with T. Please explain further. Also, what are the assumptions and limitations by introducing this kind of regularization and how do boundary values affect the framework? This needs more explanation.

Response: In this section we first extended the Gaussian process based gradient matching to the case of partial observation case. We pay attention to the treatment of the unobserved variables. After the gradient matching, we suggest the combination of L2 optimization, which minimize the difference between observation and solution under corresponding parameters, no regularization terms are included in the cost functional. Please see equation (11) in the revised manuscript.

The last step is to solve the following minimization problem,

$$\min_{\theta} \|\mathbf{x}_M(\theta) - \mathbf{y}\|_{L^2(0,T)}^2 = \min_{\theta} \sum_{i=1}^{N_2} |\mathbf{x}_M(t_i) - \mathbf{y}(t_i)|^2. \quad (11)$$

We consider initial value problems, and in Gaussian process based gradient matching there is no special treatment for the initial conditions observation since it matches the time derivatives and the equation right hand side term.

Comment #13 In Section 3 the authors reproduce the Experiments by Weng et al. As mentioned non-biological examples are missing.

Response: In the examples of Wenk et al., all the variables in a system are observable. However, here in each system one or more variable are unknown. We have supplemented a non-biological example in Section 3.5.

Comment #14 More important, I did not found any supplementary code or ready-to use examples. Was this done in Matlab? I feel in the case of this work the code and examples needs to be published, as well. Especially for a final review the code is needed!

Response: Data and relevant code for this research work are stored in GitHub: <https://github.com/yuchensufe/PartialObservation-GPGM/tree/v1.0> and have been archived within the Zenodo repository: <https://doi.org/10.5281/zenodo.4501573>

Comment #15 Page 7 Line 36... "refer FGPGM to the adapted FGPGM" this sounds odd and should be changed.

Response: Thank you for your suggestion. It has been changed as

For the Gaussian regression step for observable variables, the code published alongside Wenk et al. (2019)[25] was used. The gradient matching part should then be adapted to the partial observation case according to the diagram provided above. **In the following we refer to FGPGM as the modified fast Gaussian process gradient matching algorithm that is adapted for partial observation case.**

Comment #16 There are too much tables with confusing numbers. The authors should

consider to generate head-maps instead. (For example Table 1/Sensitivity tables)

Response: Thank you for the suggestion. We have changed them into head-maps.

Comment #17 I miss some comments regarding the running time and robustness and impact of the link function $h(y=h(x)+e)$. Also Page 13 line 33.. "Thus a least square optimization after doing FGPGM may well reduce this effect (of smoother trajectories)". Isn't the opposite the case after applying the L2 norm?

Response: In the L2 optimization step, there is no additional regularization term introduced in the cost functional (Eq.11). Only the L2 norm difference between the observation and the solution at test parameters is minimized. The introducing of Gaussian process in the gradient matching also has a denoising effect. Therefore, the trajectories after L2 optimization are not necessarily smoother than that with the Gaussian process gradient matching.

Comment #18 In the current version the discussion is very weak and needs improvement, also Page 18 Line 53 is not supported by the results. How does the dimension of the ODE system translate into computational costs and accuracy? In practice systems have more than 10 Equations and species.

Response: Thanks for the comments. In the revised manuscript, we have some discussion in Section 3.4 on the computational costs for gradient matching computation (Gaussian process based) and numerical integration (numerical integration is also involved in the L2 optimization). The main focus in this paper is the methods for parameter inference with partial observations (with unobserved variables). At present we use some small scale benchmarks, because for large systems there are many other specific issues such as the identifiability, initial guess which will be very specific to certain problems. For large scale systems, the gradient matching computation in each sampling cycle may be less time consuming than numerical integration. However, the sampling number for Gaussian process method and iteration number for optimization method need to be considered, which depends on factors such as initial guess and local minima and seems not direct to compare.

Thank you again for your helpful comments and suggestions.

Appendix B

Response Letter

Dear Professor Len Thomas,

Thank you very much for your letter and for the careful review concerning our manuscript entitled “Numerical Method for Parameter Inference of Nonlinear ODEs with Partial Observations” submitted to Royal Society Open Science. We have studied the comments carefully, which are very valuable and helpful. In the revised manuscript we supplemented some results and discussions to illustrate the performance in real situations, together with other discussions according to the comments, which we hope meet with approval.

The revisions in the manuscript and the responses to Referee’s comments are listed as follows:

Thank you very much.

Sincerely yours,

Comments to the Author(s)

The current version of the manuscript reads well, and the concept is much clearer now. Nevertheless, the authors should proofread their manuscript twice (e.g., page 6 line 23).

Response: Thank you for your careful review. We have proofread the manuscript and corrected the typos.

Comments: *I really appreciate the accessible code to support this work. In lines 46 and 53 on page 1 the authors make the point that they can deal with “partial observations” and “large noise”. I highly appreciate section 3.4 and other edits to detail this further. But still, I suggest including a better benchmark analysis. I suggest performing a proper analysis for example 3.3. I would like to see a table or AUC plot which demonstrates how the parameter error evolves with increasing noise and with exclusion of observations (sample size and complete dynamics/variables). This should be a straightforward exercise. Also, I recommend including the run-time. I am wondering how the run-time is related to the noise level.*

Response: Thank you very much for your valuable suggestions. We have carried out the analysis of parameter error dependence on latent variables and noise levels. The dependence on noise level is supplemented in Table 12 of the revised manuscript. The noise analysis is also discussed in detail for example 3.1 (Table 2-Table 9).

The dependence on latent variables are not incorporated in the manuscript due to the following considerations:

I) As shown in Table 1 below, the errors of 2nd, 3rd and 5th parameters are smaller in the case with 2 missing variables than that with 1 latent variable. However, the reconstructed states in the latter case (Figure 1 below) seems better than the 2 latent variable case (Figure 2 here). This is reasonable as the solution is a result of the parameter set in the system.

II) The results may vary if we chose different combinations of latent variables. This is related to the sensitivity of parameters and variables, which makes the problem complicated.

III) In fact, in this system part of the variables are sufficient to identify the parameters, sometimes additional observations or improper weights among the variables may cause overfitting, resulting in limited improvement of the results.

We have mentioned in the manuscript that these issues need further investigation. In order not to cause confusion, we do not show the results in the present manuscript.

In this algorithm, the run-time depends on the total sampling number we set. Therefore, the increase of noise will not affect the run time with the same sampling number. In example 3.3 the latent variables are inferred by integration which saves time for such simple system. Therefore, exclusion of observations reduces total run time (fewer calculations of probability in each sampling cycle), as shown in Table 1 below. This have been illustrated in detail in section 3.4.

Latent variable	Runtime	$ \theta_1 - \theta_1^* $	$ \theta_2 - \theta_2^* $	$ \theta_3 - \theta_3^* $	$ \theta_4 - \theta_4^* $	$ \theta_5 - \theta_5^* $
x_4	877.2s	0.0009	0.0632	0.0140	0.0068	0.0572
x_3, x_4	576.4s	0.0016	0.0603	0.0031	0.0241	0.0455
x_2, x_3, x_4	341.5s	0.0055	0.1312	0.0590	0.1316	0.0308

Table 1. Errors of inferred parameters under various latent variable numbers.

Figure 1. Inference results for example 3.3 with x_4 being latent.

Figure 2. Inference results for example 3.3 with x_3 and x_4 being latent.

Comments: *Related to the comment above I would like to see how the algorithm performs on real data. Or alternatively in cases where the noise is not Gaussian. I feel the algorithm performs well on toy examples but not on real data.*

Response: Thank you for your suggestion. To simulate real situation, we supplemented a case with abnormal measurements and fewer observable variables, in which some observed values are significantly deviated from the ground truth (Figure 12 in the revised manuscript). The results seem not obviously impacted by the errors. We have also pointed out the performance on real data need further investigation.

Comments: *In addition, the authors should investigate the case of uncertain/unknown initial state conditions, as well. If this is not possible it has to be pointed out in the discussion that the performance with real data has to be further investigated.*

Response: We have added a case with the initial condition being unknown (Figure 17 in the revised manuscript). By the present method, the system is still identifiable without initial condition of the second variable. We have also stated that the requirement of initial conditions for identifiability depends on specific systems.

Comments: *The sentence on page 6 line 16 "This approach does not depend on the observed variables." is hard to understand. The authors should consider rephrasing the sentence. It is a little bit confusing.*

Response: Thank you for pointing out that, we have rephrased it.

Thank you again for your helpful comments and suggestions.